# MYC induces CDK4/6 inhibitors resistance by promoting pRB1 degradation

Jian Ma [1,2,3,7], Lei Li [1,2,3,7], Bohan Ma[1,2,3,7], Tianjie Liu [1,2,3], Zixi Wang[1,2,3], Qi Ye[1,2,3], Yunhua Peng [4], Bin Wang[1,2,3], Yule Chen[1,2,3], Shan Xu[1,2,3], Ke Wang[1,2,3], Fabin Dang [5], Xinyang Wang[1,2,3], Zixuan Zeng[1,2,3], Yanlin Jian[1,2,3], Zhihua Ren[6], Yizeng Fan[1,2,3], Xudong Li[1,2,3], Jing Liu [1,2,3], Yang Gao[1,2,3], Wenyi Wei[5] & Lei Li [1,2,3] ✉

CDK4/6 inhibitors (CDK4/6i) show anticancer activity in certain human malignancies, such as breast cancer. However, their application to other tumor types and intrinsic resistance mechanisms are still unclear. Here, we demonstrate that MYC amplification confers resistance to CDK4/6i in bladder, prostate and breast cancer cells. Mechanistically, MYC binds to the promoter of the E3 ubiquitin ligase KLHL42 and enhances its transcription, leading to RB1 deficiency by inducing both phosphorylated and total pRB1 ubiquitination and degradation. We identify a compound that degrades MYC, A80.2HCl, which induces MYC degradation at nanomolar concentrations, restores pRB1 protein levels and re-establish sensitivity of MYC high-expressing cancer cells to CDK4/6i. The combination of CDK4/6i and A80.2HCl result in marked regression in tumor growth in vivo. Altogether, these results reveal the molecular mechanisms underlying MYC-induced resistance to CDK4/6i and suggest the utilization of the MYC degrading molecule A80.2HCl to potentiate the therapeutic efficacy of CDK4/6i.

Cancer is often defined as a disease of cell proliferation that is tightly regulated by molecular events that control the cell cycle. The cyclin-dependent kinases CDK4 and CDK6 (CDK4/6) govern progression through the early G1 phases of the cell cycle, and therefore they show promising vulnerability to cancer therapy[1]. Pharmacological inhibitors targeting CDK4/6 have shown profound effects against several solid tumors and have been approved for the treatment of hormone receptor (HR)-positive, human epidermal growth factor receptor 2 (HER2)-negative advanced or metastatic breast cancer[2–4]. However, patients with other cancer types may not benefit equally from CDK4/6i. The intrinsic resistance mechanisms of CDK4/6i and their application to additional cancer types warrant further investigation.

A number of putative mechanisms that confer resistance to CDK4/6 inhibition have been identified, such as the loss of retinoblastoma protein RB1 (via direct deletion or *RB1* mutations), CDK6 amplification, activating alterations in AKT1, RAS, AURKA, CCNE2, ERBB2 and FGFR2, and loss of estrogen receptor (ER) expression[5–10]. Among these mechanisms, the loss of normal RB1 function is the most frequently observed change in cells resistant to CDK4/6i, largely because of the canonical role played by pRB1 as a major CDK4/6 substrate[1,10–14]. pRB1 negatively regulates cell cycle progression by binding to and sequestering E2F transcription factors, thereby preventing entry into S phase of the cell cycle[15,16]. pRB1 phosphorylation via CDK4/6 leads to the release of E2F transcription factors, which

[1]Department of Urology, The First Affiliated Hospital of Xi'an Jiaotong University, Xi'an 710061, China. [2]Key Laboratory of Environment and Genes Related to Diseases, Ministry of Education, Xi'an 710061, China. [3]Key Laboratory for Tumor Precision Medicine of Shaanxi Province, The First Affiliated Hospital of Xi'an Jiaotong University, Xi'an 710061, China. [4]Center for Mitochondrial Biology and Medicine, The Key Laboratory of Biomedical Information Engineering of Ministry of Education, School of Life Science and Technology, Xi'an Jiaotong University, Xi'an 710049, China. [5]Department of Pathology, Beth Israel Deaconess Medical Center, Harvard Medical School, Boston, MA 02115, USA. [6]Kintor Parmaceutical, Inc, Suzhou 215123, China. [7]These authors contributed equally: Jian Ma, Lei Li, Bohan Ma. ✉e-mail: lilydr@163.com

ultimately drive cell cycle progression[17–20]. RB1 loss is one of the most fundamental events in many cancers[21,22], such as breast cancer[23] and prostate cancer[24,25]. While the genetic loss of *RB1* has been extensively studied, the effect of the inactivation of the pRB1 protein at the posttranslational level remains poorly understood.

MYC is one of the most widely investigated oncoproteins that regulates many cellular processes and contributes to tumorigenesis and therapeutic resistance in several different cancer types[26–29]. The MYC family includes the ubiquitously expressed MYC, the less broadly distributed MYCN, and the extensively studied MYCL family members[30]. MYC functions as a universal amplifier of transcription through its interaction with numerous factors and complexes[31]. The MYC gene is overexpressed in as many as 70% of human cancers[26,32], which makes MYC an attractive theoretical target for cancer treatment[33]. Silencing MYC in multiple tumor models leads to tumor regression associated with remodeling of the tumor microenvironment[28,29]. Although multiple efforts have been made, it is still a big challenge to target MYC with clinical-grade small molecules, especially at the protein level[34,35].

In the present study, we show that high MYC expression drives resistance to CDK4/6i by directly activating the transcription of the E3 ubiquitin ligase KLHL42, which promotes pRB1 ubiquitination and degradation and thus leads to pRB1 protein deficiency. In addition, we characterize a molecule that degrades MYC, A80.2HCl, to abolish MYC when applied at nanomolar levels, rescues pRB1 protein activity, and diminishes MYC-dependent CDK4/6i resistance. Moreover, the combination of CDK4/6i and MYC-degrading molecule A80.2HCl shows an additive effect on killing tumor cells both in vitro and in vivo.

## Results

### High MYC expression drives resistance to CDK4/6i by masking pRB1

The cooccurrence or mutual exclusivity of cancer genome alterations usually indicates potential interactions between these altered gene functions or their downstream pathways[36–39]. To characterize the regulatory mechanism of RB1 in-depth, we performed large-scale genomic analyses of The Cancer Genome Atlas (TCGA) database and found that *MYC* amplification and *RB1* deletion were found to be almost mutually exclusive in multiple cancer types[40–42] (Supplementary Fig. 1a), indicating potential cross regulation between the RB1 and MYC pathways. We hypothesize that those *MYC* amplification cancer cells may undergo RB1 loss in mRNA or protein level without *RB1* deletion. To test it, we measured the expression of RB1 and MYC in bladder, prostate and breast cancer cell lines (Supplementary Fig. 1b) and found that pRB1 protein levels were negatively correlated with MYC expression (Supplementary Fig. 1c). Given that RB1 loss is a key reason for CDK4/6i resistance[1,11,12], we hypothesized that high MYC expression might affect the treatment sensitivity of cells to CDK4/6i in part by suppressing pRB1 activity. To this end, we measured the viability of several bladder, prostate and breast cancer cells treated with the CDK4/6 inhibitor palbociclib. We found that palbociclib markedly inhibited the proliferation of cells expressing low levels of MYC and formed fewer and smaller colonies, while cells expressing high levels of MYC showed only a slight decrease in the number and size of the colonies formed (Supplementary Fig. 2a). A dose–survival assay demonstrated that the IC50 of palbociclib in low-MYC-expressing cells was much lower than that in high-MYC-expressing cells (Supplementary Fig. 2b). Moreover, the IC50 of palbociclib was negatively correlated with MYC protein levels but positively correlated with pRB1 protein levels (Supplementary Fig. 2c, d). To further explore the potential of palbociclib to induce a response in patients, we conducted a pharmacological test of palbociclib effectiveness on mini-PDX models derived from breast cancer tissues in which *MYC* was expressed at different levels. Notably, breast cancer tissues with *MYC* amplification showed palbociclib resistance (Fig. 1a).

We further investigated the transcriptional landscape between MYC and RB1. As expected, palbociclib treatment resulted in a significant decrease in gene expression in normal MYC-expressing T24 cells but exerted little effect in high-MYC-expressing UMUC14 cells (Fig. 1b, c). Moreover, after knocking down MYC in T24 cells, in addition to that of MYC target genes, the expression of E2F target genes was downregulated (Fig. 1d, e). It has been reported that E2F1 is a transcriptional target of MYC that promotes cell cycle progression[43,44]. However, MYC knockdown led to only a slightly decrease in E2F1 mRNA levels and exerted minimal influence on E2F1 protein levels in T24 and UMUC14 cells (Supplementary Fig. 2e, f), indicating that the transcriptional landscape change after MYC knockdown was not dependent on the regulation of E2F1 expression.

To recapitulate MYC amplification in patients, we overexpressed MYC in T24 bladder cancer cells, C4-2 prostate cancer cells and MDA-MB-231 breast cancer cells. Overexpression of MYC exerted minimal influence on E2F1 protein levels but reduced pRB1 protein abundance in all three cell lines (Fig. 1f). Moreover, although palbociclib treatment inhibited the colony formation in T24, C4-2 and MDA-MB-231 cells carrying an empty vector (EV), the effect of palbociclib was largely suppressed in MYC-overexpressing cells (Fig. 1g, Supplementary Fig. 2g). We further showed that MYC overexpression conferred palbociclib resistance onto T24 xenograft tumors in mice (Fig. 1h, i). Thus, our data suggest that high MYC expression drives resistance to CDK4/6i by reducing pRB1 protein abundance in cancer cells.

### High MYC expression reduces pRB1 abundance via proteasomal degradation

As previous studies have shown that MYC is a master transcription factor[31], we doubted that MYC could transcriptionally repress the expression of RB1. However, we found that ectopic expression of MYC decreased the pRB1 protein expression level in a dose-dependent manner and that this effect was completely abrogated by the proteasome inhibitor MG132 (Fig. 2a). Furthermore, overexpression of MYC did not change the mRNA level of *RB1* (Fig. 2b). Moreover, knocking down MYC with two independent shRNAs increased the level of endogenous pRB1 proteins but exert no overt effect on RB1 mRNA expression in either T24 or UMUC14 cells (Fig. 2c, d). We also noticed that the level β-TrCP, which we had previously reported to be an E3 ligase of pRB1 in breast cancer cells[45], was not changed after MYC knockdown, indicating that in this experimental setting, MYC may have regulated RB1 protein levels independent of the β-TrCP level. We further observed that knocking down MYC prolonged the half-life of endogenous pRB1 protein in T24 and UMUC14 cells (Fig. 2e–h). Indeed, knockdown of MYC markedly compromised the polyubiquitination of the pRB1 protein (Fig. 2i). To examine the effect of MYC on pRB1 protein levels in patient specimens, we analyzed MYC and pRB1 protein levels in a tissue microarray (TMA) containing a cohort of bladder cancer samples (*n* = 40) using immunohistochemistry (IHC) and found that pRB1 expression was negatively correlated with MYC expression (Fig. 2j, k). Together, these findings provide direct evidence that high MYC expression plays a causal role in repressing pRB1 abundance via proteasomal degradation.

### The E3 ubiquitin ligase KLHL42 interacts with pRB1 and induces pRB1 proteasomal degradation

To identify the potential E3 ubiquitin ligase that targets pRB1 for degradation, we performed LC–MS analysis to identify proteins interacting with pRB1 in the context of cells with or without MYC overexpression (Fig. 3a). We demonstrated that the E3 ubiquitin ligase KLHL42[46–48] showed increased binding to pRB1 after MYC was overexpressed (Fig. 3a). Through a co-IP assay, we confirmed that ectopically expressed KLHL42 interacted with pRB1 in T24 cells (Supplementary Fig. 3a). An interaction between endogenous pRB1 with KLHL42 proteins instead of other ubiquitin-associated proteins

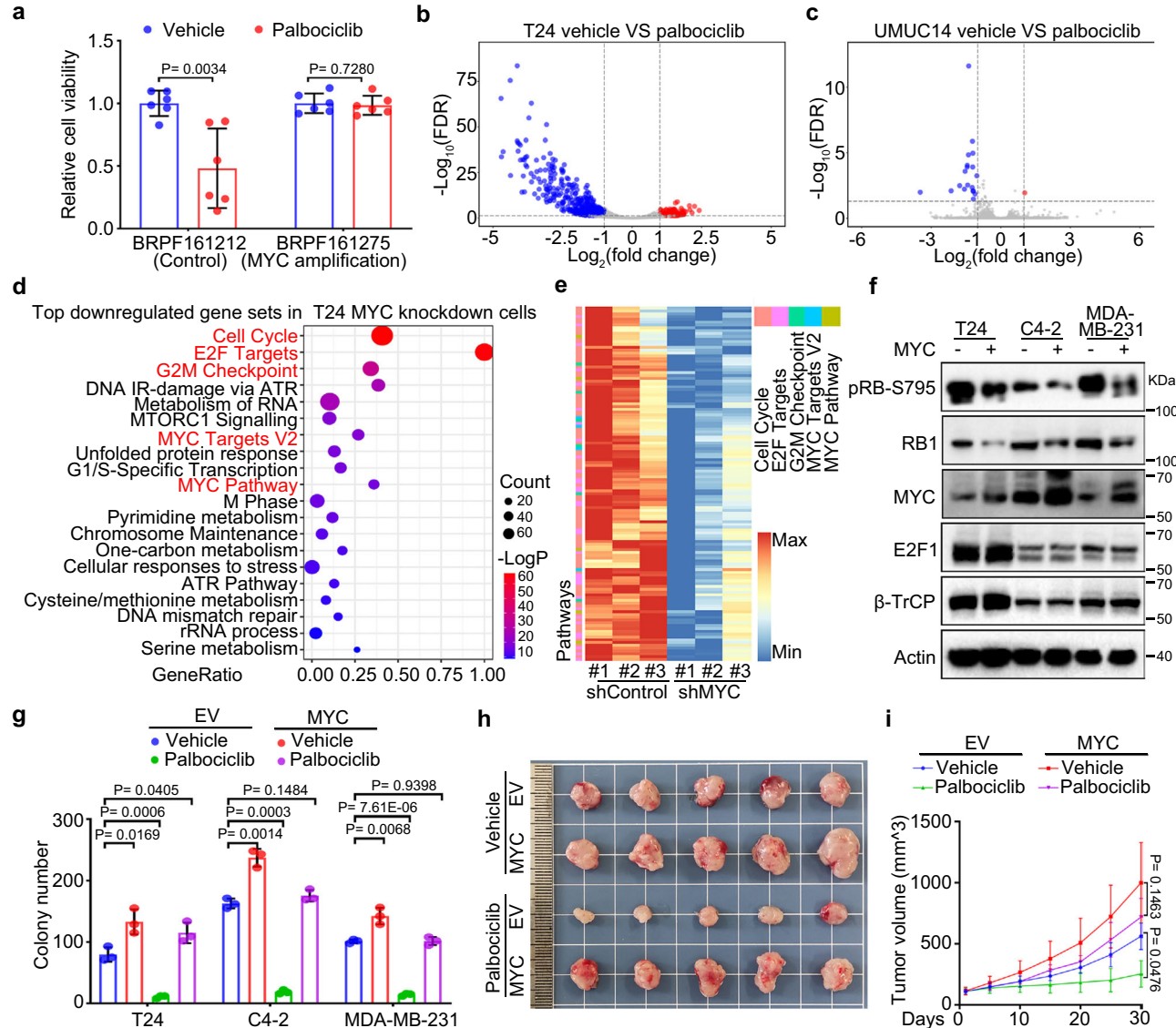

**Fig. 1 | High MYC expression drives resistance to CDK4/6i by masking pRB1.**
**a** Pharmacological tests of palbociclib using mini-PDX models[72,73], palbociclib sensitivity results for control and MYC amplified tumors. Data were shown as the mean ± SD of six mice ($n = 6$). Two-tailed unpaired Student's $t$-test. $P$ values based on the order of appearance: 0.0034 and 0.7280. Volcano plots showing differentially regulated genes by RNA-seq analysis from CDK4/6i treated T24 (**b**) and UMUC14 (**c**) cells. **d** Control or MYC-knockdown T24 cells were harvested for RNA-seq analysis and pathway analysis. Differentially regulated genes with more than 2-fold change were included in this pathway analysis. Color represents the level of significance, and $P$ value adjusted. Data is analyzed by standard accumulative hypergeometric statistical test. **e** Heatmap showing the differential expression of the indicated pathway genes downregulated after MYC knockdown in T24 cells. **f** Control or MYC-overexpressing T24, C4-2, and MDA-MB-231 cells were harvested

for western blotting with the indicated antibodies. **g** Quantification of colony formation of control or MYC-overexpressing T24, C4-2, and MDA-MB-231 cells treated with vehicle or palbociclib. Data were shown as the mean ± SD of three independent experiments ($n = 3$). Two-tailed unpaired Student's $t$-test. $P$ values based on the order of appearance: 0.0169, 0.0006 and 0.0405; 0.0014, 0.0003 and 0.1484; 0.0068, 7.61E-06 and 0.9398. **h, i** Control or MYC-overexpressing T24 cells were injected s.c. into the right flank of NSG mice. Tumor volume was measured at the indicated time points. Tumors were harvested and photographed on day 30 (**h**). In **i**, data were shown as the mean ± SD of five mice ($n = 5$). Two-way ANOVA (two-sided). $P$ values based on the order of appearance: 0.0476 and 0.1463. Source data are provided in this paper. Similar results for **f** panel were obtained in three independent experiments.

identified from the screening was detected in T24 cells overexpressing MYC (Fig. 3b). In the reversed co-IP assay, we demonstrated that KLHL42 also showed specific binding to pRB1 but not other cell cycle factors such as RBL1/p107, RBL2/p130 or CDK4/6 (Fig. 3c). To identify the region(s) in pRB1 critical for binding to KLHL42, we constructed different domains in pRB1 as reported previously[49] (Fig. 3d). Both the co-IP and Glutathione S-transferase (GST) pull-down assay using GST-tagged recombinant pRB1 truncations confirmed the direct interaction between pRB1 and KLHL42 and indicated the N-terminal domain of pRB1 was crucial for pRB1 binding with KLHL42 (Fig. 3e and

Supplementary Fig. 3b). Similarly, we constructed KLHL42 deletion plasmids based on its functional domains (Fig. 3f) and found that the Kelch domain, which had been previously reported to be a substrate-binding domain[50–52], was critical for pRB1 binding (Fig. 3g). Thus, we identified a physiological interaction between the E3 ligase KLHL42 and pRB1, which was augmented when MYC expression was amplified.

We then sought to examine whether KLHL42 regulates pRB1 protein levels. We found that pRB1 protein levels were decreased after ectopic expression of wild-type (WT) KLHL42 but were restored by the proteasome inhibitor MG132 (Supplementary Fig. 3c). However,

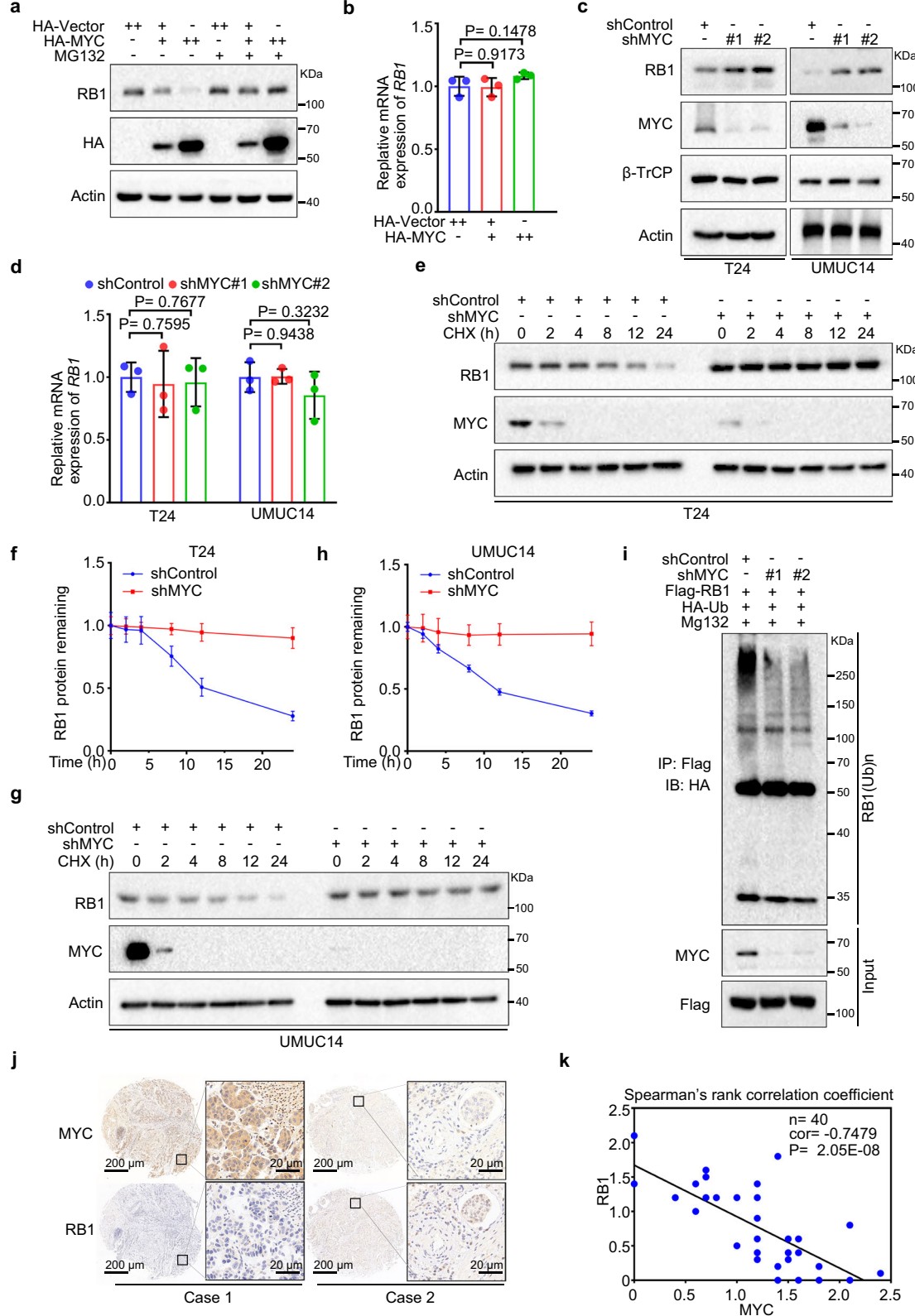

KLHL42 Kelch domain or BTB domain deletion mutants failed to decrease pRB1 protein levels (Supplementary Fig. 3c). Moreover, both the hypo-phosphorylated[19] (all 15 Ser/Thr Cdk acceptor sites changed to Ala) and hyper-phosphorylated (S245D) pRB1 mutants were degraded by KLHL42 expression (Supplementary Fig. 3d, e). Consistent with the findings after MYC knockdown, KLHL42 knockdown increased endogenous total and phosphorylated pRB1 protein instead of other

pocket proteins, without changing the RB1 mRNA level in both T24 and UMUC14 cells (Fig. 3h, i and Supplementary Fig. 3f, g). We further showed that KLHL42 expression augmented pRB1 polyubiquitination (Fig. 3j). On the other hand, KLHL42 knockdown attenuated pRB1 polyubiquitination in 293 T cells (Fig. 3k). Thus, we revealed that KLHL42 expression clearly shortened the pRB1 protein half-life (Supplementary Fig. 3h, i). In contrast, knocking down KLHL42 prolonged

**Fig. 2 | High MYC expression reduces pRB1 abundance via proteasomal degradation.** 293 T cells were transfected with an HA vector (EV) or HA-tagged MYC for 24 h. Cells were treated with or without MG132 for 6 h and harvested for western blotting (**a**) and RT–qPCR (**b**). In **b**, data were shown as the mean ± SD of three independent experiments (n = 3). Two-tailed unpaired Student's t-test. P values based on the order of appearance: 0.9173 and 0.1478. Control or MYC-knockdown T24 and UMUC14 cells were harvested for western blotting (**c**) and RT–qPCR analyses (**d**). In **d**, Data were shown as the mean ± SD of three independent experiments (n = 3). Two-tailed unpaired Student's t-test. P values based on the order of appearance: 0.7595, 0.7677, 0.9438 and 0.3232. Control or MYC-knockdown T24 cells were treated with 200 μg/μl CHX for Western blotting (**e**). Protein bands were quantified in **f**. In **f**, data were shown as the mean ± SD of three independent experiments (n = 3). Control or MYC-knockdown UMUC14 cells were treated with 200 μg/μl CHX for Western blotting (**g**). Protein bands were quantified in **h**. In **h**, data were shown as the mean ± SD of three independent experiments (n = 3). **i** Control or MYC-knockdown 293 T cells were transfected with the indicated plasmids and harvested for IP under denaturing conditions and subjected to immunoblotting. **j** Representative images of IHC analysis with anti-MYC and anti-RB1 antibodies on TMA (n = 40 TMA elements) tissue sections. Scale bar in 10 X fields: 200 μm; Scale bar in 40 X fields: 20 μm. **k** Correlation analysis of the IHC staining of MYC and pRB1 proteins in bladder cancer patient specimens. Two-sided Spearman correlation coefficient, P = 2.05E-8. Source data are provided in this paper. Similar results for (**a**, **c**, **e**, **g** and **i**) panels were obtained in three independent experiments.

the half-life of the endogenous pRB1 protein in T24 and UMUC14 cells (Fig. 3l, m, Supplementary Fig. 3j, k). Consistently, knocking down MYC or KLHL42 restored pRB1 protein level in T24, C42 and MDA-MB-231 cell lines (Supplementary Fig. 3l). Since pRB1 plays an important role in cell cycle, we further investigated how pRB1 and KLHL42 is regulated during cell cycle. Cell cycle analysis reaved that both the pRB1 and KLHL42 protein abundance fluctuated during the cell cycle. pRB1 protein peaked in the G2/M phase but slightly declined in the early G1/S phase as previously reported[53,54], revealing an inverse correlation with KLHL42 (Supplementary Fig. 3m, n). Furthermore, knockdown of MYC decreased the KLHL42 protein level but increased the pRB1 protein and phosphorylation level in all cell cycle phases (Supplementary Fig. 3m, n). Thus, we demonstrated that KLHL42-induced pRB1 degradation is independent of phosphorylation status of pRB1 and is persistent throughout cell cycle.

We further analyzed KLHL42 protein levels using the same TMA presented in Fig. 2j. Similarly, the expression levels of KLHL42 and pRB1 in patient specimens were negatively correlated (Fig. 3n and Supplementary Fig. 4a). Moreover, a TCGA database analysis showed that *KLHL42* mRNA was amplified in pancancer contexts, especially in breast cancer, bladder cancer and prostate cancer (Supplementary Fig. 4b), and was significantly correlated with MYC protein levels in bladder cancer (Supplementary Fig. 4c). Collectively, these results suggest that KLHL42 is a bona fide E3 ligase that modulates pRB1 protein ubiquitination and thus decreases pRB1 stability in cancers.

### KLHL42 is a transcriptional target of MYC that mediates CDK4/6i resistance

As described above, MYC and KLHL42 both reduce pRB1 abundance via proteasomal degradation, and KLHL42 markedly interacted with pRB1 after MYC was overexpressed (Figs. 2, 3). We sought to examine the relationship between MYC and KLHL42. By performing a meta-analysis of published ChIP-seq datasets, we found an MYC-binding peak at the promoter of the *KLHL42* gene in H2171 and SW1271 human lung cancer cells[55,56] (Fig. 4a). A ChIP–qPCR analysis confirmed the substantial enrichment of MYC at the promoter of the *KLHL42* gene in T24 and UMUC14 cells (Fig. 4b). To experimentally verify that *KLHL42* is a transcriptional target of MYC, we knocked down MYC in T24 and UMUC14 cells and found that KLHL42 expression was decreased at both the mRNA and protein levels in these two cell lines (Fig. 4c, d). We further analyzed the expression of Myc, Klhl42 and Rb1 in prostate specimens from Pbscn-cre and *High Myc* transgenic mice[57]. While a histological analysis indicated that Rb1 expression was markedly reduced in both normal and tumor specimens from the *High Myc* mice compared with the Pbsn-cre mice, the expression of Myc and Klhl42 was significantly increased in the *High Myc* mice (Fig. 4e, f). These data indicate that KLHL42 is a direct MYC target that promotes pRB1 degradation.

To further assess the effect of KLHL42 on MYC-mediated CDK4/6i resistance, we measured the viability of T24 and UMUC14 cells with MYC or KLHL42 knocked down either independently or together. KLHL42 knockdown markedly inhibited the proliferation of CDK4/6i-

treated cells, which also formed fewer and smaller colonies regardless of MYC expression level, whereas KLHL42 knockdown in control cells (vehicle-treated) only slightly decreased the size and number of the colonies formed (Fig. 4g–i). Similarly, silencing MYC and KLHL42 individually or together resulted in a decreased IC50 for palbociclib compared with that in control cell lines (Fig. 4j, k). Therefore, we demonstrated that KLHL42 is a MYC downstream target that induces pRB1 degradation and mediates CDK4/6i resistance. Knocking down either KLHL42 or MYC-sensitized tumor cells to CDK4/6i treatment.

### Identification of A80.2HCl as a MYC-degrading molecule

Given that MYC is frequently overexpressed in multiple cancer types[26,32] and that we had demonstrated that MYC overexpression led to CDK4/6i resistance, development of drugs that efficiently target MYC is urgently needed. As previously reported, compounds featuring an imide skeleton, such as pomalidomide and thalidomide, have demonstrated a strong affinity for E3 ligase cereblon (CRBN)[58]. In pursuit of identifying potential drug candidates for MYC degradation, we initiated the construction of a molecular library comprising 177 distinct imide-based compounds (Supplementary Data 3). Our selection of HL60, a MYC drug-sensitive cancer cell line, served as the model system for this screening process. Among the numerous candidates, we identified and further evaluated those showing significant inhibitory effects on cancer cell proliferation (Supplementary Data 4 and 5). Encouragingly, we discovered a compound, A80.2HCl, with the ability to completely degrade the MYC protein at a remarkably low concentration of 10 nM (Supplementary Data 5, and patent application WO 2023/116835 A1). Using in vitro screening, drug metabolism, pharmacokinetics methods and in vivo validation, we found that A80.2HCl specifically binds to GSPT1 and MYC and subsequently recruits CRBN for degradation (Fig. 5a). The further characterizations of A80.2HCl with high-resolution mass spectrometry and NMR analysis are available in the Supplementary Information (Supplementary Figs. 5–8) and patent application (WO 2023/116835 A1). Moreover, we demonstrated that A80.2HCl is predicted to form interactions with both CRBN and MYC by molecular docking analysis. Specifically, A80.2HCl predominantly binds to a domain consisting of amino acids at positions 324 to 340, a location that aligns with the previously reported binding sites of CRBN for pomalidomide and lenalidomide[59] (Supplementary Fig. 9a). A80.2HCl interacts with key residues in MYC, including Glu 432, Arg 436 and Arg 439, which situated within the MAX interface (Fig. 5b). To precisely determine the binding affinity between A80.2HCl and MYC protein, we employed Isothermal Titration Calorimetry (ITC) assay and revealed that A80.2HCl bound to MYC at 145 nM in vitro (Fig. 5c). The Homogeneous Time-Resolved Fluorescence (HTRF) analysis further confirmed the binding between A80.2HCl and CRBN (Fig. 5d). Consistently, co-IP assay showed that A80.2HCl effectively induced the binding of CRBN to MYC in T24 cells (Fig. 5e).

To further investigate the efficiency of A80.2HCl action on the MYC, KLHL42 and pRB1 axes identified in our study, we treated bladder cancer, prostate cancer and breast cancer cell lines with increasing

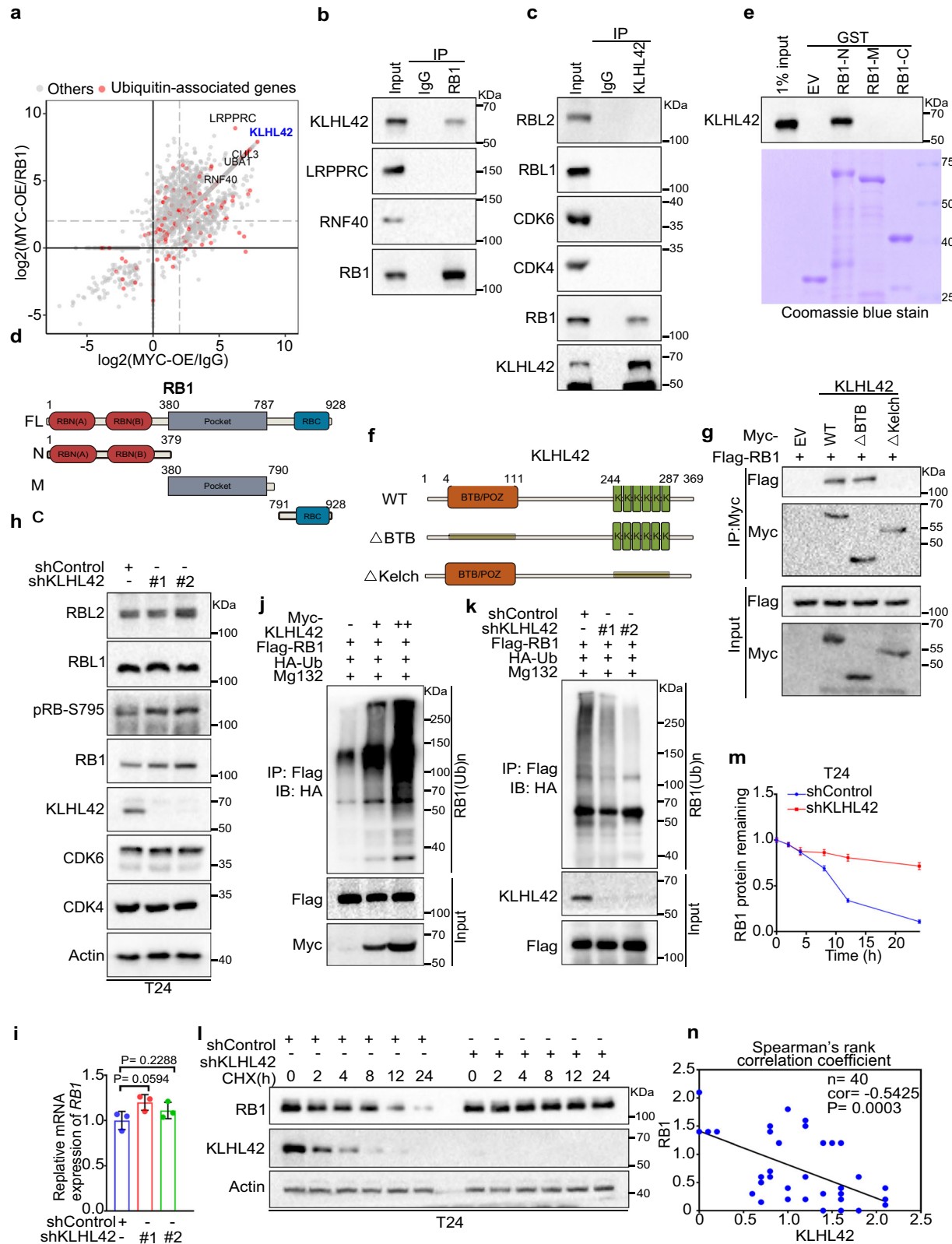

doses of A80.2HCl. As expected, MYC was degraded when A80.2HCl was administered at 2 to 50 nM in different cell lines, and this effect was accompanied by a notable KLHL42 level decrease and pRB1 level increase (Fig. 5f, Supplementary Fig. 9b, d). Consistent with MYC knockdown, A80.2HCl decreased KLHL42 mRNA levels, while RB1 mRNA levels remained unchanged (Supplementary Fig. 9c). Moreover, both MLN4924 and MG132 rescued A80.2HCl-mediated MYC

degradation, indicating that A80.2HCl reduced MYC abundance via proteasomal degradation (Supplementary Fig. 9e, f). Indeed, A80.2HCl prolonged the pRB1 protein half-life, similar to the effect of MYC knockdown (Supplementary Fig. 9g–j). Furthermore, we performed an RNA-seq assay and found that the A80.2HCl treatment markedly attenuated the transcription signatures of both MYC and E2F target genes (Fig. 5g, h), further supporting the on-target effect of A80.2HCl.

**Fig. 3 | The E3 ubiquitin ligase KLHL42 interacts with pRB1 and induces pRB1 proteasomal degradation. a** IP-MS ratio plot showing the normalized log2(MYC-OE/RB1) against log2(MYC-OE/IgG). Ubiquitin-associated genes are highlighted in pink. **b, c** Coimmunoprecipitated endogenous RB and KLHL42 proteins in T24 cells overexpressing MYC. **d** Schematic diagram depicting a set of truncated pRB1 constructs used in this study. **e** Western blot analysis of KLHL42 proteins in T24 WCL pulled down by GST or GST-RB recombinant proteins. **f** Schematic diagram depicting the KLHL42 deletion mutants used in this study. **g** 293 T cells transfected with the indicated plasmids were harvested for immunoprecipitation (IP) under denaturing conditions and subjected to immunoblotting. Control or KLHL42 knockdown T24 cells were harvested for western blotting (**h**) and RT–qPCR analyses (**i**). In **i**, data were shown as the mean ± SD of three independent experiments ($n = 3$). Two-tailed unpaired Student's $t$-test. $P$ values based on the order of

appearance: 0.0594, and 0.2288. **j** 293 T cells were transfected with increased Myc-KLHL42 in combination with Flag-RB1 and HA-Ub and harvested for IP under denaturing conditions and subjected to immunoblotting. **k** Control or KLHL42-knockdown 293 T cells were transfected with the indicated plasmids and harvested for IP under denaturing conditions and subjected to immunoblotting. Control or KLHL42-knockdown T24 cells were treated with 200 μg/μl CHX for Western blotting (**l**). Protein bands were quantified in **m**. In **m**, data were shown as the mean ± SD of three independent experiments ($n = 3$). **n** Correlation analysis of IHC staining of KLHL42 and pRB1 proteins in bladder cancer patient specimens. Two-sided spearman correlation coefficient, $P = 0.0003$. Source data are provided in this paper. Similar results for (**b, c, e, g, h, j, k** and **l**) panels were obtained in three independent experiments.

However, knocking out CRBN completely abolished A80.2HCl-induced MYC degradation, indicating that A80.2HCl likely functions in a CRBN-dependent manner (Fig. 5i, Supplementary Fig. 9k).

We then sought to determine the sensitivity of tumor cells to A80.2HCl. We demonstrated that A80.2HCl significantly inhibited colony formation in all the studied bladder cancer, prostate cancer and breast cancer cell lines (Fig. 5j, k, Supplementary Fig. 10a, b). Notably, A80.2HCl showed a favorable IC50 range, from 12 to 78 nM, in all cell lines (Fig. 5l, Supplementary Fig. 10c, d). However, MYC knockdown in bladder cancer cells led to significant resistance to A80.2HCl effects (Fig. 5j-l), indicating that the inhibitory effect of A80.2HCl likely depended on MYC expression. In addition, A80.2HCl treatment significantly inhibited both T24 and UMUC14 xenograft tumor growth in mice but exerted little effect on MYC-knockdown xenograft tumors (Fig. 5m, n, Supplementary Fig. 10e, f). Moreover, an IHC analysis of the xenograft tumors showed a marked decrease in MYC protein levels in the A80.2HCl-treated groups (Supplementary Fig. 10g, h). To evaluate the pharmacokinetic (PK) properties of the A80.2HCl drug, A80.2HCl was first labeled with rhodamine B, and then, we conducted an in vivo mouse pharmacokinetics assay after A80.2HCl orally administered. As shown in Fig. 5o, A80.2HCl obviously accumulated within the tumor and was eliminated from organs after 36 h. We further conducted a preliminary toxicity assessment of A80 in BALB/c mice and revealed that A80.2HCl showed no significant toxicity, as evidenced by the absence of adverse effects on body weight, liver and kidney function or other major organs (Supplementary Fig. 11). In summary, these results established that A80.2HCl as a molecule that competitively degraded MYC and as a potential therapeutic target for cancer treatment.

## A80.2HCl potentiates the therapeutic efficacy of CDK4/6 inhibitors

As demonstrated above, knocking down MYC or KLHL42 overcame CDK4/6i resistance in multiple cancer cells (Fig. 4h–k), and A80.2HCl effectively degraded MYC (Fig. 5). We sought to determine the efficiency of A80.2HCl administered in combination with palbociclib. To this end, we examined transcriptional changes in bladder cancer cells treated with A80.2HCl and palbociclib individually or together by RNA-seq analysis. Intriguingly, the combination treatment of A80.2HCl and palbociclib induced a more profound downregulation of both MYC and E2F downstream target levels compared with that induced by either single drug treatment (Fig. 6a). Notably, a subset of 1609 genes was downregulated only by the A80.2HCl and palbociclib combination treatment (Fig. 6b). We further demonstrated that this group of genes was enriched with MYC and E2F targets (Fig. 6c), indicating that the combination of A80.2HCl with palbociclib arrested more MYC and E2F targets than either single agent administered alone.

We then sought to evaluate the therapeutic effect of this combination strategy in vitro and in vivo. A80.2HCl-treated cells showed a significantly decreased IC50 for palbociclib compared to vehicle-treated cells (Fig. 6d, e). Moreover, cotreatment of T24 and UMUC14 cells with A80.2HCl and palbociclib resulted in a much greater

inhibitory effect on colony formation (Fig. 6f, g). Using compuSyn analysis[60], we confirmed the combination of A80.2HCl with palbociclib synergistically killed T24 and UMUC14 cells (Supplementary Fig. 12a, b). We further examined the effect of MYC plus CDK4/6 inhibitors in KLHL42 overexpressed cells. As expected KLHL42 overexpression abolished the combined effect of MYC plus CDK4/6 inhibitors in both colony formation and cell viability assays (Supplementary Fig. 12c–g). Moreover, the combination treatment of A80.2HCl plus palbociclib did not show significant improvement compared with A80.2HCl treatment alone in RB1 deficient BT549 cells (Supplementary Fig. 12h-j). These results indicated that pRB1 expression is necessary for both MYC and CDK4/6 inhibitors. We further examined this effect using other CDK4/6 inhibitors. Consistently, the efficacy of both abemaciclib and ribociclib in colony formation and cell viability was markedly enhanced as palbociclib did upon addition of A80.2HCl (Supplementary Fig. 13a–c).

Although CDK4/6 inhibitors have been approved for clinical therapy, acquired resistance to CDK4/6 inhibitors has emerged. To validate whether MYC amplification is involved in acquired resistance to CDK4/6 inhibitors, we generated the CDK4/6i resistant cell lines, which was obtained by continuously being treated with CDK4/6i (Supplementary Fig. 14a). Intriguingly, we found that the pRB1 protein levels were dramatically reduced, while MYC and KLHL42 protein abundance was accumulated in resistant daughter cells (Supplementary Fig. 14a). Notably, employment of A80.2HCl overcomes the resistance of CDK4/6i caused by MYC and KLHL42 protein accumulation (Supplementary Fig. 14b). Addition of A80.2HCl re-sensitized the resistant cancer cells to CDK4/6i treatment implicated by the enhanced cell cycle gene suppression and colony formation inhibition (Supplementary Fig. 14c–e). Thus, our results provide intriguing cellular evidence pointing to MYC as an attractive target for preventing or overcoming CDK4/6i resistance in cancer cells.

Similar to the in vitro results, the addition of A80.2HCl markedly enhanced the efficacy of palbociclib in retarding T24 and UMUC14 tumor growth in vivo (Fig. 6h, Supplementary Fig. 15a–c). Furthermore, we evaluated the efficacy of the combination treatment of A80.2HCl and palbociclib in a mini-PDX model of MYC amplification (Fig. 6i). The addition of A80.2HCl markedly enhanced the efficacy of palbociclib in vivo. In summary, these results established that the MYC-degrading molecule A80.2HCl potentiates the therapeutic efficacy of CDK4/6i in inhibiting tumor growth, thereby providing an effective cancer treatment drug target for overcoming CDK4/6i resistance.

## Discussion

RB1 was one of the first tumor suppressors to be identified and is a master regulator of cell cycle progression[61,62]. Cancer genomic studies have indicated that RB1 loss has been frequently detected in multiple cancer types[21–25]. Because RB1 is a major downstream effector of CDK4/6 signaling, RB1 loss in cancer cells contributes intrinsic resistance to CDK4/6i[1,10–14]. However, as most studies have focused on genetic *RB1* alterations, little is known about the

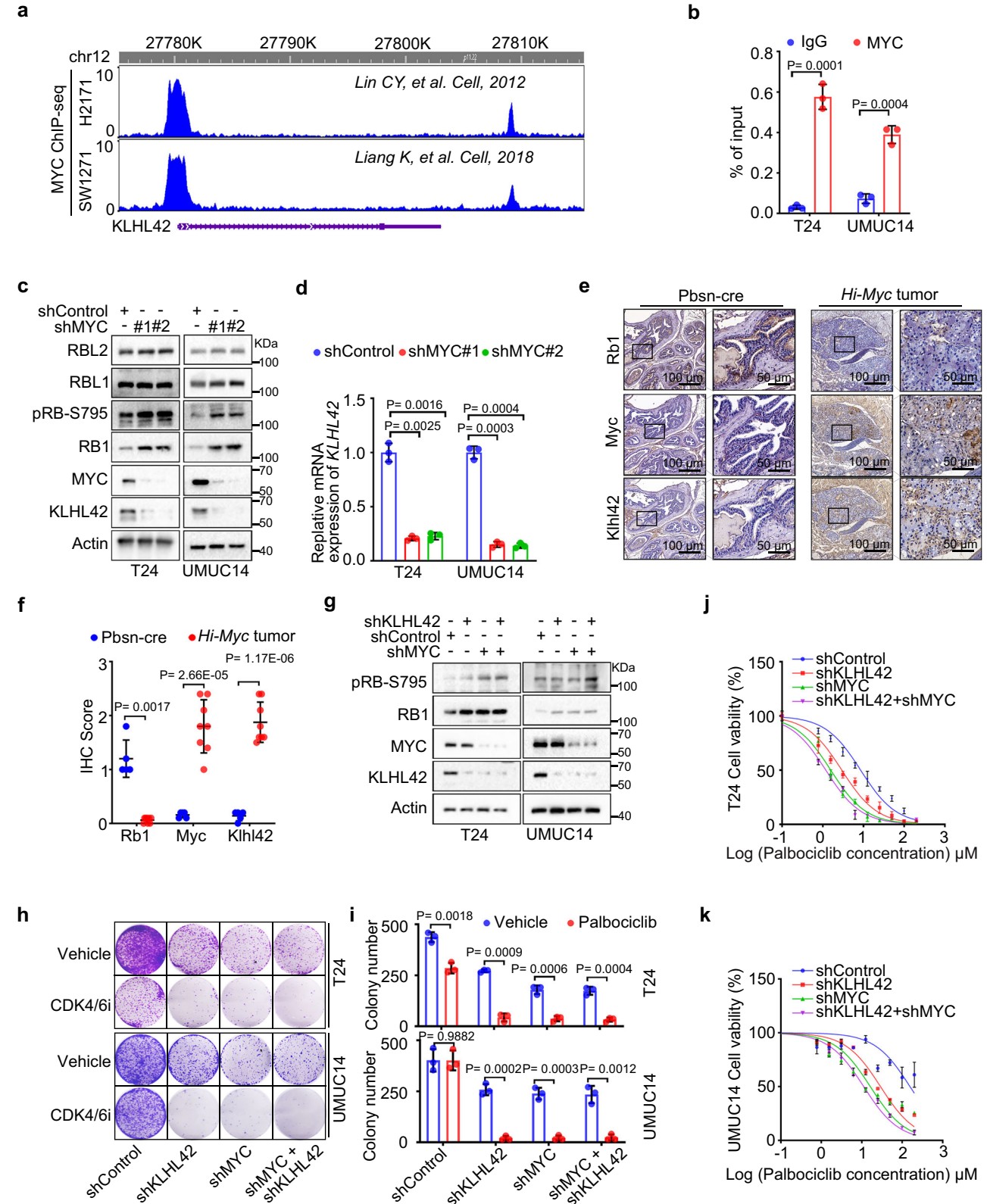

posttranslational modification of pRB1, except for phosphorylation[61]. In the present study, we demonstrated that the E3 ubiquitin ligase KLHL42 targets pRB1 for degradation. Intriguingly, KLHL42 is a transcriptional target of MYC, which links RB1 loss with MYC amplification or overexpression. We further demonstrated that MYC amplification and RB1 loss led to similar effect on the transcriptional landscape associated with CDK4/6i resistance. Moreover, data from

mouse tumors and human bladder cancer tissues confirmed the repression of pRB1 protein abundance mediated by MYC and KLHL42. Analyses of the TCGA database further supported our findings suggesting that *MYC* amplification and *RB1* deletion were found to be almost mutually exclusive in multiple cancer types[40–42] (Supplementary Fig. 1a). These findings demonstrate that, in addition to the genetic alterations of RB1, MYC amplification and induction of

**Fig. 4 | KLHL42 is a transcriptional target of MYC that mediates CDK4/6i resistance. a** Screenshot of the UCSC genome browser showing ChIP-seq signal profiles of MYC in the KLHL42 gene locus in different human cell lines, as previously reported[55,56]. **b** ChIP–qPCR analysis of MYC binding at the promotor of the KLHL42 gene in T24 and UMUC14 cells. Data were shown as the mean ± SD of three independent experiments (n = 3). Two-tailed unpaired Student's t-test. P values based on the order of appearance: 0.0001, and 0.0004. Control or MYC-knockdown T24 and UMUC14 cells were harvested for western blotting (**c**) and RT–qPCR analyses (**d**). In **d**, data were shown as the mean ± SD of three independent experiments (n = 3). Two-tailed unpaired Student's t-test. P values based on the order of appearance: 0.0025, 0.0016, 0.0003, and 0.0004. **e, f** Representative images of IHC analysis with anti-RB1, anti-Myc and anti-KLHL42 antibodies on FFPE samples of prostate specimens from Pbsn-Cre and High Myc transgenic mice and the quantitative data of Rb1, Myc and Klhl42 staining were shown in **f**. Scale bar in 10 X fields: 200 μm; Scale bar in 40 X fields: 20 μm. In **f**, data were shown as the mean ± SD (Pbsn-cre n = 5, Hi-Myc tumor n = 8). Two-tailed unpaired Student's t-test. P values based on the order of appearance: 0.0017, 2.66E-05, and 1.17E−06. **g** T24 and UMUC14 cells infected with the indicated lentivirus expressing shRNAs were harvested for western blotting. T24 and UMUC14 cells infected with the indicated lentivirus expressing shRNAs treated with vehicle or palbociclib were harvested for colony formation assay (**h**). In **i**, data were shown as the mean ± SD of three independent experiments (n = 3). Two-tailed unpaired Student's t-test. P values based on the order of appearance: 0.0018, 0.0009, 0.0006 and 0.0004; 0.9882, 0.0002, 0.0003 and 0.0012. T24 (**j**) and UMUC14 (**k**) cells infected with the indicated lentivirus expressing shRNAs treated with palbociclib were harvested for the cell viability assay. Data were shown as the mean ± SEM of three independent experiments (n = 3). Source data are provided in this paper. Similar results for **c** and **g** panels were obtained in three independent experiments.

KLHL42 play important roles in the inactivation of pRB1 and drive CDK4/6i resistance.

The MYC oncoprotein is always considered an undruggable target due to its intrinsically disordered nature and lack of a binding pocket[63]. Several strategies have been explored to directly or indirectly inhibit MYC, including inhibition of binding with its co-factors[64,65], preventing MYC expression[66] and inducing synthetic lethality via MYC overexpression[67,68]. However, no compound has yet been found suitable for clinical testing. Recently, targeted protein degradation (TPD) has become a therapeutic modality and is paving the way for targeting previously undruggable proteins[69]. Furthermore, small molecules promoting MYC degradation have been suggested to be promising treatments for MYC-driven cancers[70]. The molecule, A80.2HCl, generated in our study, can degrade MYC when applied at nanomolar concentrations to multiple cancer cells and exhibits a strong inhibitory effect on tumor growth. While our study focuses mainly on bladder, prostate and breast cancers, more studies are warranted to characterize the anticancer function of A80.2HCl in other cancer types.

In summary, our work reveals a previously unrecognized role of MYC in driving CDK4/6i resistance by disrupting pRB1 protein stability. High MYC expression activates the E3 ligase KLHL42, which is responsible for pRB1 ubiquitination and degradation and subsequently confers resistance to CDK4/6i to cancer cells. Moreover, we identified a MYC-degrading molecule, A80.2HCl, that efficiently reduces MYC protein levels and overcomes CDK4/6i resistance (Fig. 7), shedding light on strategies for the expanded use of CDK4/6i in cancer treatment.

## Methods

### Cell culture, stable transfectants, and transfection
293T, T24, 235J, UMUC3, UMUC14, Vcap, MDA-MB-231, MCF7 and T47D cells were cultured in DMEM medium supplemented with 10% FBS. PC-3, C4-2 and 22RV1 cells were maintained in RPMI 1640 medium supplemented with 10% FBS. All cell lines were kept in a 37 °C incubator at 5% CO2. Cells were routinely checked for mycoplasma infection and tested negative. The identity and purity of cell lines were validated using the short tandem repeat method by the vendors. Transfection was performed using PEI (Polysciences) or Lipofectamine 2000 (Thermo Fisher Scientific) according to the manufacturer's instructions. For lentiviral infection, 293T cells were transfected with packaging vectors (pMD2.G and psPAX2) and MYC shRNA, KLHL42 shRNA, pLKO plasmids, then the virus-containing supernatant was collected 48 h after transfection. Indicated cancer cells were infected with virus-containing supernatant in the presence of polybrene (4 μg/ml) and were then selected in growth medium containing 1.5 μg/ml puromycin at least three days. Cycloheximide (CHX) assays were performed as described previously[71].

### Antibody information
Primary antibodies used were RB (Cell Signaling Technology, # 9309, 1:1000), RB (Cell Signaling Technology, # 9313, 1:1000), E2F1 (Proteintech, #66515-1-Ig, 1:1000), Phospho-Rb (Ser795) (Cell Signaling Technology, # 9301S, 1:1000), RBL1 (Proteintech, #13354-1-AP, 1:1000), RBL2 (Proteintech, #27251-1-AP, 1:1000), Cyclin B1 (ABclonal, # A19037, 1: 1000), RNF40 (ABclonal, # A6443, 1: 1000), LRPPRC (ABclonal, # A3365, 1: 1000), CDK4 (ABclonal, # A11136, 1: 1000), CDK6 (ABclonal, # A0106, 1: 1000), KATNA1 (ABclonal, # A16491, 1: 1000), c-MYC (ABclonal, # A1309, 1: 1000), KLHL42 (Invitrogen, # PAS-54292, 1: 500), Vinculin (Santa Cruz, # sc-73614, 1:1000), β-Actin (Cell Signaling Technology, # 4970, 1:1000), Myc (Santa Cruz, # sc-40, 1:1000), HA (Cell Signaling Technology, # 3724, 1:1000), Flag (MBL, #M185-7, 1:1000), Rabbit IgG (ABclonal, # AS014, 1:3000), Mouse IgG (ABclonal, # AS003, 1:3000).

### Immunoblotting and immunoprecipitation
Briefly, cells were harvested and lysed with IP buffer (50 mM Tris−HCl, pH 7.4; 150 mM NaCl; and 0.1% NP-40) containing protease inhibitors (Complete Mini, Roche) and phosphatase inhibitors (cocktail set I and II, Calbiochem), followed by centrifugation at 12,000 × g for 10 min at 4 °C. The supernatant was collected and quantified by BCA protein quantification assay. Protein samples were prepared with 5x SDS loading buffer (250 mM Tris−HCl, pH 6.8; 10% SDS; 25 mM β-mercaptoethanol; 30% glycerol; and 0.05% bromophenol blue) and boiled for 5 min. Equal amounts of protein samples were then subjected to SDS–PAGE analysis and transferred to nitrocellulose membranes. The membranes were blocked with 5% milk for 1 h at room temperature and incubated with primary antibody at 4 °C overnight. The next day, the membranes were washed three times with 1x TBST (20 mM Tris, 100 mM NaCl, and 0.1% Tween-20) and incubated with secondary antibodies for 1 h at room temperature. The protein bands were visualized by using ECL western blotting substrate (Bio−Rad) and visualized by a western film processor.

For immunoprecipitation analysis, cells were harvested and lysed by IP buffer (50 mM Tris−HCl, pH 7.4; 150 mM NaCl; and 0.1% NP-40) containing protease inhibitors (Complete Mini, Roche) and phosphatase inhibitors (cocktail set I and II, Calbiochem) on ice for more than 30 min. The cell lysate was centrifuged for 15 min at 12,000 × g at 4 °C, and the supernatant was incubated with primary antibody-conjugated protein A/G beads (Thermo Fisher Scientific) or HA-/Flag-conjugated agarose beads (Sigma–Aldrich) rotating at 4 °C overnight. The next day, the beads were washed at least four times with IP buffer on ice and then subjected to western blot analysis.

### In vivo ubiquitination assays
293 T cells were transfected with HA-tagged ubiquitin and the indicated plasmids. Twenty-four hours after transfection, cells were treated with 20 μM MG132 for 6 h and lysed with IP buffer (50 mM Tris−HCl, pH 7.4; 150 mM NaCl; and 0.1% NP-40) containing protease inhibitors (Complete Mini, Roche) and phosphatase inhibitors (cocktail set I and II, Calbiochem) on ice for more than 10 min. The lysate was sonicated and centrifuged for 15 min at 12,000 × g at 4 °C, and the

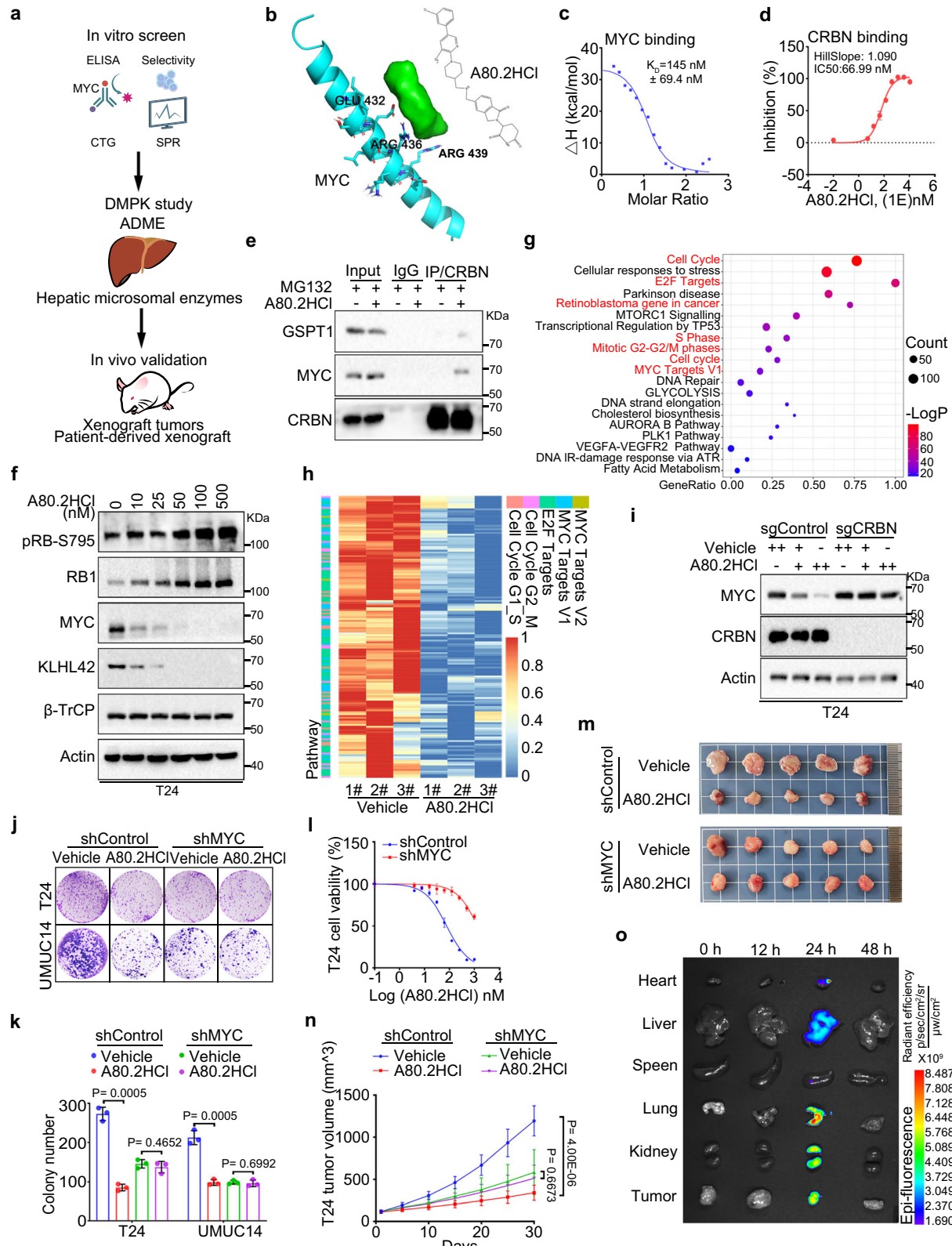

supernatant was incubated with Flag-conjugated agarose beads (Sigma–Aldrich) while rotating at 4 °C overnight. The next day, the bound beads were washed four times with IP buffer on ice, and then, the proteins were subjected to western blotting analysis. The ubiquitinated form of pRB1 was measured by western blotting using an anti-HA antibody.

**Total RNA extraction and quantitative real-time polymerase chain reaction (qRT–PCR) analysis**

Total RNA was extracted using an RNAfast200 kit (Fastagen), and reverse transcription was performed using PrimeScript RT Master Mix (RR036A, TAKARA) following the manufacturer's instructions. Quantitative real-time PCR was performed using technical triplicates and

**Fig. 5 | Identification of A80.2HCl as a MYC-degrading molecule. a** Workflow of in vitro and in vivo screening performed to identify A80.2HCl. **b** Docking simulation cartoon illustrating that A80.2HCl bound to MYC (PDB ID: 1A93). **c** The binding affinity between the A80.2HCl drug and MYC was measured by ITC. **d** The binding affinity of the A80.2HCl drug and CRBN was measured by Homogeneous Time-Resolved Fluorescence (HTRF). Data were shown as mean ± SD of three independent experiments (*n* = 2). **e** Coimmunoprecipitated endogenous CRBN, GSPT1 and MYC proteins from T24 cells. **f** T24 cells treated with increased quantities of A80.2HCl were harvested for western blotting. **g** T24 cells treated with vehicle or A80.2HCl were harvested for RNA-seq analysis and pathway analysis. Differentially regulated genes with more than 2-fold change were included in this pathway analysis. Data is analyzed by standard accumulative hypergeometric statistical test. *P* value adjusted. **h** Heatmap showing the differential expression of the indicated pathway genes suppressed by A80.2HCl treatment in T24 cells. **i** CRBN^WT and CRBN^KO T24 cells treated with vehicle and A80.2HCl were harvested for western

blotting. Control or MYC-knockdown T24 and UMUC14 cells treated with A80.2HCl were harvested for colony formation assay (**j**). In **k**, data were shown as the mean ± SD of three independent experiments (*n* = 3). Two-tailed unpaired Student's *t*-test. *P* values based on the order of appearance: 0.0005, 0.4652, 0.0005 and 0.6992. **l** Control or MYC-knockdown T24 cells treated with A80.2HCl were harvested for cell viability assay. Data were shown as the mean ± SEM of three independent experiments (*n* = 3). **m, n** Control or MYC-knockdown T24 cells were injected s.c. into the right flank of NSG mice and treated with the indicated drugs. Tumor was measured at the indicated time points and harvested on day 30 (**m**). In **n**, data were shown as the mean ± SD of five mice (*n* = 5). Two-way ANOVA (two-sided). *P* values based on the order of appearance: <0.0001 and 0.6673. **o** Biodistribution of A80.2HCl in T24 xenograft models. Source data are provided in this paper. Similar results for (**e**, **f** and **i**) panels were obtained in three independent experiments.

biological replicates with *2×TSINGKE® Master qPCR Mix (SYBR Green I) (TSINGKE) and a CFX96 detection system (Bio–Rad)*. 18S was used for normalization, and the 2 − ΔΔCt method was used for quantitative analysis. Primer sequences are listed in Supplementary Data 1.

### Drug response test of breast cancer mini patient-derived xenograft (mini-PDX) models

To rapidly test drug efficacy in vivo, we established mini-PDX models according to the protocols reported in previously published papers[72,73]. Tumor cells derived from breast cancer tissues with different expression levels of the MYC protein were harvested and digested into single cells. The cells were then filled into OncoVee® capsules (LIDE Biotech, Shanghai, China). Each capsule contained ~2000 cells. The capsules were implanted subcutaneously via a small skin incision, with 3 capsules per mouse (5-week-old female nu/nu mouse). Mice bearing the Mini-PDX capsules were treated with the appropriate control or drugs (vehicle or Palbociclib and A80.2HCl). Palbociclib and A80.2HCl were administered orally in a single administration (daily [qd] x 1) for 7 continuous days at doses of 100 mg/kg and 6 mg/kg body weight, respectively. All these drugs were prepared in DMSO, PEG300 and Tween-80 solutions. Vehicle controls were an isometric 0.5% HPMC and a 0.2% Tween-80 solution, and the vehicle treatment was performed the same way as drug treatment. After all the capsules were removed from the mice, the number of proliferating tumor cells in each capsule was counted using a CellTiter Glo Luminescent Cell Viability Assay kit (G7571, Promega, Madison, WI, US). The tumor cell growth inhibition rate was calculated using the published formula.

### Experimental therapy of xenograft mouse models

For T24 xenograft studies, T24 ($1 \times 10^6$) cells were suspended in 100 μl of serum-free DMEM, mixed with Matrigel (Corning, 354234; 1:1) and then injected into the flanks of male nude mice. The mice were randomly grouped (ten mice for each group), and the treatment was started when the tumor size reached 100–150 mm³. Tumor size was measured every 5 d with a caliper, and the tumor volume was determined based on the formula L × W2 × 0.5, where L is the longest diameter and W is the shortest diameter. The allowed maximal tumor size is 2 cm in any direction based on the institutional tumor production policies and none of the tumors exceeded this size at any point. For the A80.2HCl treatment assay, when the tumor volume reached 100–150 mm³, the xenografted mice were randomly assigned to groups and treated daily with vehicle (0.5% methyl cellulose), palbociclib (100 mg/kg), A80.2HCL (6 mg/kg), or a palbociclib + A80.2HCl combination by gastric gavage. Tumor volume and weight were measured as mentioned above. The experiments with mice were conducted according to protocols approved by the Rules for Animal Experiments published by the Chinese Government and approved by the Institutional Animal Care and Use Committee of Xi'an Jiaotong University.

### Immunohistochemical analysis

Bladder cancer tissue microarray (TMA) slides and tumor specimens from mice were stained with antibodies against RB1, MYC and KLHL42 via standard immunohistochemistry procedures. Images were acquired using a Leica SCN400 microscope. Semiquantification of protein expression was performed on the basis of the following scoring criteria: percentage of positively stained cells (%) and staining intensity (0, no staining; 1, week staining; 2 moderate staining; 3 strong staining) were measured, and then, the values were multiplied to yield a score ranging from 0 to 3. To maintain consistency, the same qualified pathologist interpreted all the IHC data.

Formalin-fixed, paraffin-embedded (FFPE) samples of prostate specimens from High Myc transgenic mice used for the IHC analysis in our study were kindly provided by Professor Jiangang Long (School of Life Science and Technology and Frontier Institute of Science and Technology, Xi'an Jiaotong University, Xi'an, Shaanxi, China).

### Analysis of A80.2HCl binding to CRBN or MYC by molecular docking

The AlphaFold-predicted full-length structure of CRBN was used for A80.2HCl docking. To prepare the MYC structure (PDB ID: 1A93)[74] for the simulations, we first removed water molecules, ions, and co-crystallized ligands, followed by an energy minimization step to optimize its conformation. As for the A80.2HCl ligand, its 2D structure, represented in SMILES notation, was converted into 3D coordinates and energetically minimized for optimal structure. The grid box dimensions were set at 30 Å × 30 Å × 30 Å, with a grid point distance of 0.05 nm, allowing for comprehensive coverage of the protein domains and enabling free molecular movement. Docking simulations were performed using the DockThor online software with an exhaustiveness level of 1000 to ensure a thorough exploration of ligand-protein binding configurations. The DockTScore scoring function was employed for evaluating ligand-protein interactions. We also validated DockThor's reliability by conducting redocking experiments with the co-crystallized ligand and assessing the root mean square deviation (RMSD) in comparison with the crystallographic pose, with acceptable RMSD values serving as a reliability criterion. Subsequent post-docking analysis facilitated a detailed examination of binding interactions. PyMOL was used to visualize and analyze the results of our docking simulations.

### Cell viability assays

Cells (3000 cells/well) for each condition were plated in 96-well plates and cultured in 100 μl of the indicated medium containing 10% serum. After 24 h, cells were treated with various concentrations of compounds in 100 μl of medium or left untreated for 24 to 48 h, and cell viability was assessed using methyl thiazol tetrazolium (MTT, Sigma Aldrich) according to the manufacturer's instructions. The absorbance at a wavelength of 570 nm was read using an Epoch Microplate

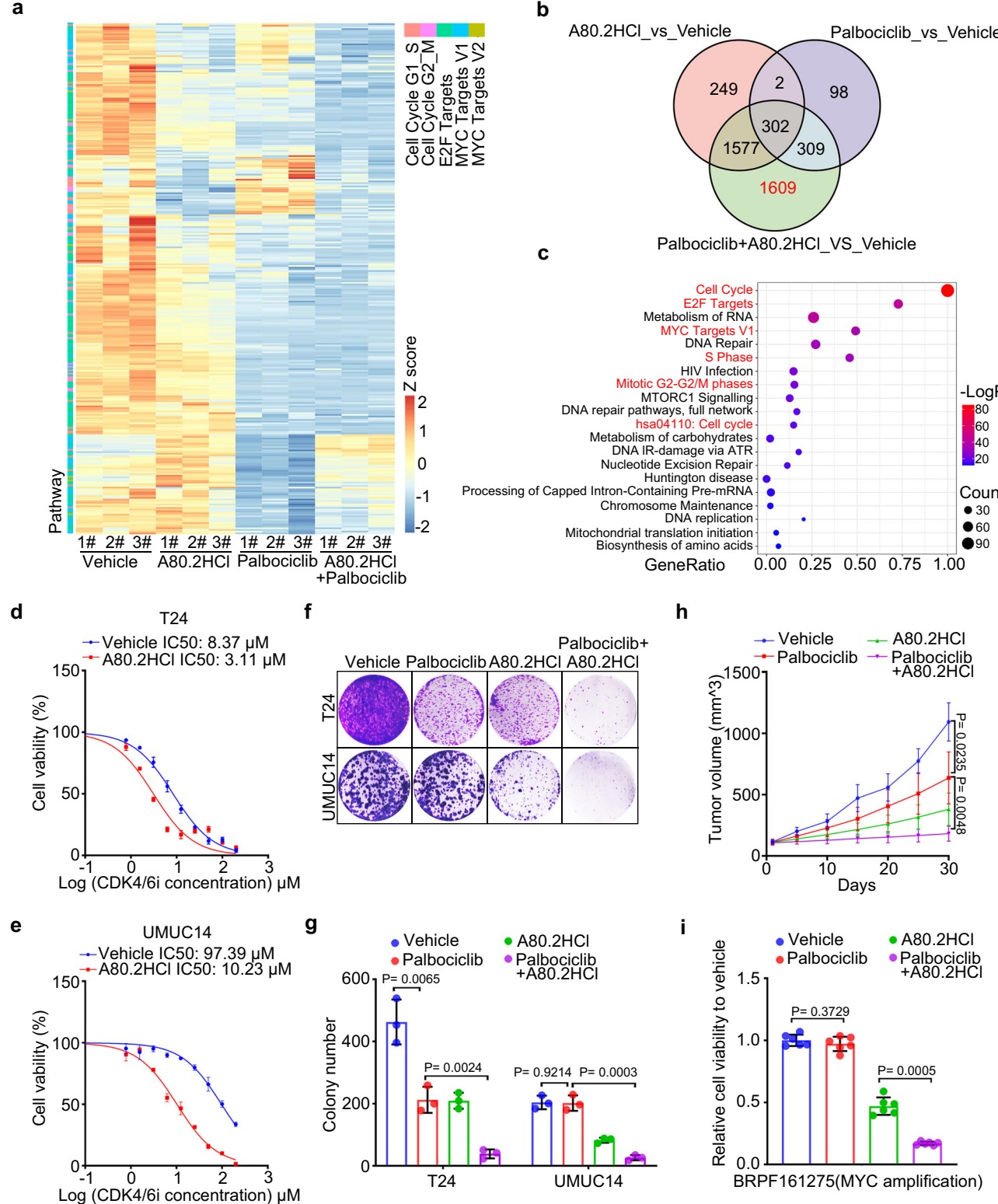

Spectrophotometer (BioTek). All cell viability experiments were conducted in triplicate, and the average value represents the value of a single biological replicate.

### Colony formation assays

Cells were seeded in six-well plates (1000 cells/well) in the indicated medium and were cultured for 1–2 weeks depending on the size of the colony. Then, the cells were fixed with 4% paraformaldehyde for 15 min and stained with crystal violet (0.5% w/v) for 30 min before the

colonies were counted. The colonies were gently washed with running water. The number of colonies with more than 50 cells was recorded.

### Mass spectrometry analysis

Liquid chromatography–mass spectrometry (LC–MS) analysis was performed as described previously[75]. 293 T cells were transfected with Flag-tagged RB1 with or without MYC. After 48 h, the cells were lysed with IP buffer and immunoprecipitated with Flag-conjugated agarose beads (Sigma–Aldrich). The bound proteins were eluted

**Fig. 6 | A80.2HCl potentiates the therapeutic efficacy of CDK4/6i. a** T24 cells treated with the indicated drugs were harvested for RNA-seq analysis, and heatmap showing the differential expression of the indicated pathway genes. **b** Venn diagram showing the overlapping downregulated genes after the indicated treatment of T24 cells. **c** Pathway analysis of 1609 genes downregulated after A80.2HCl + palbociclib treatment. Differentially regulated genes with more than 2-fold change were included in this pathway analysis. Color represents the level of significance, and *P* value adjusted. Data is analyzed by standard accumulative hypergeometric statistical. T24 (**d**) and UMUC14 (**e**) cells treated with the indicated drugs were harvested for cell viability assays. Data were shown as the mean ± SEM of three independent experiments (*n* = 3). T24 and UMUC14 cells treated with the indicated drugs were harvested for colony formation assay (**f**). In **g**, data were shown as the mean ± SD of three independent experiments (*n* = 3). Two-tailed unpaired Student's *t*-test. *P* values based on the order of appearance: 0.0065, 0.0024, 0.9214 and 0.0003. **h** T24 cells were injected s.c. into the right flank of NSG mice, which were treated with the indicated drugs. Tumor volume was measured at the indicated time points. Data were shown as the mean ± SD of five mice (*n* = 5). Two-way ANOVA (two-sided). *P* values based on the order of appearance: 0.0235 and 0.0048. **i** Drug sensitivity analysis with mini-PDX models of breast cancer tissues with *MYC* amplification. Data were shown as the mean ± SD of six mice (*n* = 6). Two-tailed unpaired Student's *t*-test. *P* values based on the order of appearance: 0.3279, and 0.0005. Source data are provided in this paper.

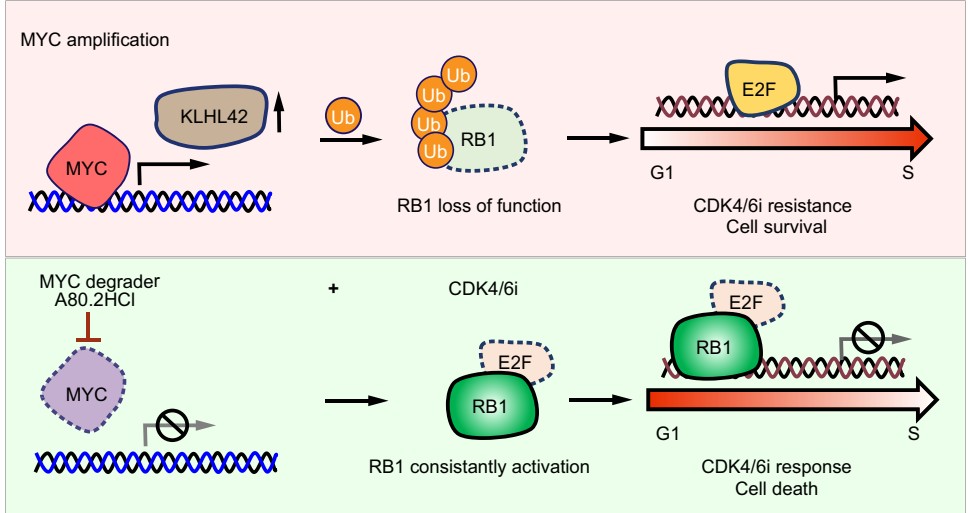

**Fig. 7 | A schematic illustration of the working model.** MYC amplification drives CDK4/6i resistance by activating an E3 ligase KLHL42, which is responsible for pRB1 ubiquitination and degradation, and subsequently confers resistance to CDK4/6i. A MYC degrader A80.2HCl that can efficiently reduce MYC protein level protects pRB1 and overcomes CDK4/6i resistance. The combination of A80.2HCl and CDK4/6i sheds light on potentially strategies for cancer treatment.

with 2% SDS and digested overnight with sequencing-grade modified trypsin (Promega, PRV5111). The obtained peptides were analyzed using a nanoflow EASY-nLC 1200 system (Thermo Fisher Scientific, Odense, Denmark) coupled to an Orbitrap Exploris480 mass spectrometer (Thermo Fisher Scientific, Bremen, Germany). The results were processed with the UniProt human protein database (75,004 entries, download on 07-01-2020) using Protein Discoverer (Version 2.4.1.15, Thermo Fisher Scientific) and Mascot (Version 2.7.0, Matrix Science).

### RNA-seq and data analysis

T24 and UMUC14 cells were treated with vehicle, palbociclib (1 µM), A80.2HCl (10 nM), or palbociclib + A80.2HCl for 24 h or infected with lentivirus expressing control or MYC-specific shRNAs followed by puromycin selection for 48 h. Total RNA was isolated from cells using TRIzol reagent (Invitrogen). High-quality (Agilent Bioanalyzer RIN > 7.0) total RNA was employed for the preparation of sequencing libraries using an Illumina TruSeq Stranded Total RNA/Ribo-Zero Sample Prep Kit. A total of 500–1000 ng of riboRNA-depleted total RNA was fragmented by RNase III treatment at 37 °C for 10–18 min, and RNase III was inactivated at 65 °C for 10 min. Size selection (50- to 150-bp fragments) was performed using the FlashPAGE denaturing PAGE-fractionator (Thermo Fisher Scientific) before ethanol precipitation overnight. The resulting RNA was directionally ligated, reverse-transcribed and treated with RNase H. A differential expression analysis was performed using the DESeq2 (v1.30.1) Bioconductor package. After adjustments via Benjamini and Hochberg's approach for controlling the false discovery rate, the genes meeting the criteria of a *P* value < 0.05, an adjusted *P* value < 0.05 and a fold change >=1.5 were considered to be differentially expressed genes.

### Chromatin immunoprecipitation (ChIP) assay

Cell lysate was sonicated and subjected to immunoprecipitation using anti-Myc antibody or nonspecific IgG. After extensive wash, immunoprecipitated DNA was amplified by real-time PCR. Sequence information for ChIP primers is provided in Supplementary Data 2.

### Quantification and statistical analysis

Graphs were generated using GraphPad Prism 8 project (GraphPad, Inc.) or Microsoft Office Excel 2010. Differences between groups were compared by *t*-tests or two-way ANOVA by GraphPad Prism 8 project for Statistical Computing. Spearman's rank correlation was used to calculate the correlation between MYC/RB1 and the MYC/KLHL42 staining index in bladder cancer TMAs. *P* < 0.05 was the criterion used to represent a significant difference. No statistical method was used to predetermine sample size.

### Reporting summary

Further information on research design is available in the Nature Portfolio Reporting Summary linked to this article.

## Data availability

All data associated with this study are present in the paper or the Supplementary Materials. The mass spectrometry proteomics raw data

have been deposited to the ProteomeXchange Consortium via the iProX partner repository with the dataset identifier PXD037479. The published MYC structure used in this study can be found in the Protein Data Bank under accession codes: 1A93[74]. Raw sequencing data have been deposited in the National Center for Biotechnology Information Sequence Read Archive under the BioSample accession SAMN31422159 and BioProject accession PRJNA893398. Further information and requests for resources and reagents should be directed to and will be fulfilled by the lead contact, Lei Li (lilydr@163.com). Source data are provided with this paper.

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

## Acknowledgements

This work was supported, in part, by grants from the National Key R&D Program of China (2023YFC3404104 to L.L.) and the National Natural Science Foundation of China (81925028 to L.L., 82230097 to L.L., 82173037 to J.M. and 82273106 to X.L.).

## Author contributions

J.M., L.L. and B.M. have contributed equally to this work. J.M. and L.L. conceived the study. J.M., L.L., T.L., Z.W., Q.Y., Y.P., B.W., Y.C., S.X., K.W., X.W., Z.Z., Y.J., B.M., Z.R. and Y.F. performed experiments, data collection, and analysis. X.L., F.D., J.L. and Y.G. provided conceptual advice. L.L., W.W. and J.M. supervised the study and wrote the manuscript.

## Competing interests

W.W. is a co-founder and consultant for the ReKindle Therapeutics. Z.R. is the VP of Kintor Pharmaceutical, Inc. Other authors declare no competing interests.

### Ethics

The study protocol of clinical samples for bladder cancer tissue microarray slides was approved by the Institutional Ethics Committee of Shanghai Outdo Biotech Company. All patients provided written informed consent to participate in the study. The mini PDX model protocol was approved by the Institutional Ethics Committee of Shanghai LIDE Biotech.
