## [Peer Review File · Nature Communications]

REVIEWER COMMENTS

Reviewer #1 - drug development, MYC degraders (Remarks to the Author):

In this report, authors first identified MYC as a key to confer cancer cell resistant to CDK4/6 inhibitors. They then found E3 ubiquitin ligase KLHL42 as a transcriptional target of MYC, which induced RB1 degradation. Further, they generated a MYC degrader, A80.2HCl and showed that it worked well to restore RB1 level and reestablished the sensitivity of cancer cells to CDK4/6i. Finally, they demonstrated that combination of CDK4/6i and A80.2HCl could generated better therapeutic effects. The experiments are well planned and performed. The results are clear and supporting the conclusions.

Major comments

1. The identification of A80.2HCl is interesting. The use of this MYC degrader is critical for this paper. However, the information of A80.2HCl is very limited. Is it a chemical or other type of molecule? There is no structure of A80.2HCl provided. Also, it is unclear how it binds to GSPT1 and MYC. Figure 5b is not clear enough and lack of detailed description of the binding information.
2. The basic information regarding the potential toxicity of A80.2HCl needs to be provided. Did it lead to body weight loss and ALT/AST increase in animal studies?
3. In Fig. 5o, A80.2HCl was shown to accumulate within the tumor. Is there any explanation for this tumor-centered accumulation ?

Reviewer #2 - Rb in cancer (Remarks to the Author):

In this manuscript, Ma et al suggest an inverse correlation between MYC and pRB expression (but see below), which confers resistance to the CDK4/6 inhibitor, palbociclib. They subsequently show that MYC induces the Cullin-3-Binding protein KLHL42, which induces ubiquitination and degradation of pRB. Inhibition of MYC reduces KLHL42 expression, increases pRB expression and induces sensitivity to the CDK4/6 inhibitor palbociclib both in vitro and in xenograft assays. The analysis is to the large part well executed and logical, and the results are both interesting and are potentially important therapeutically. However, there are several issues that are of great concern as outlined below each of which can affect the interpretation of results. In addition, the description of the mass spec analysis and the identification of MYC degrader, A80.2HCl, is insufficient - and must be revised.

Major issues:

1. In Supplementary Fig. 1b - the authors show by western blots an inverse correlation between RB and MYC expression. Several lines such as BT549 express no pRb with very high MYC expression - and the assumption/interpretation is that MYC induces degradation of RB via KLHL42.

However, at least some if not all of these lines with no discernable pRB expression including BT549 have deletions/truncations in the RB gene – and must not be included in this analysis.

(BT549, RB1 c.265_607del343; MDAMB468, RB1 c.265_2787del2523; DU4475, RB1 c.1_2787del2787; and MDAMB436, RB1 c.607_608ins107)

DU145 - RB1 Hom c.2143A>T

DO the other lines have intact RB1 gene? If MYC or KLHL42 is knocked-down/out in these cells – is pRB expression restored? If not – this figure is misleading.

Moreover – this figure shows high expression of MYC plus loss of RB1 (due to deletion/mutations), which contradicts Supplemental Fig. 1a.

2. For Fig 3a – shows FLAG-tagged RB1 immunoprecipitation from 293T cells followed by mass spectrometry analysis to identify RB1-interacting proteins –

The authors should provide all the results from this analysis as supplementary excel files as well as Volcano plots with top binders - so one can evaluate what other relevant factors were identified relative to KLHL42.

RB protein tends to precipitate and the IP experiments in Fig 3c should carefully control for this with additional antibodies to other E3 ligases or factors that do not bind pRB as negative control.

3. Does KLHL42 degrade other pocket proteins – i.e. p107 or p130?

Does KLHL42 degrade other factors that control the cell cycle that may affect CDK4/6 response?

It was shown that KLHL42 (Cul3/Klhdc5 E3 Ligase) controls microtubule density through its interaction and ubiquitination of p60/katanin (PMID: 19261606). Could this effect on p60/katanin indirectly affect pRB phosphorylation/degradation? For example, if KLHL42 inhibits cells at G2/M – could this lead to pRB hyper-phosphorylation and ubiquitination by other E3 ligases? Can the authors rule out this possibility?

4. The description of how A80.2HCl was identified is essentially missing in the Methods or text and impossible to understand or follow. The authors should explain exactly how they identified this drug – define GSPT1, CRBN and ITC - and the entire rationale.

What other drugs have been identified in this screen ? (excel sheet)

What's the structure of A80.2HCl?

Is it related to other drugs with similar MYC degradation activity?

e.g. "A Selective Small-Molecule c-Myc Degradator Potently Regresses Lethal c-Myc Overexpressing Tumors" Adv Sci 2022.

5. Does A80.2HCl have 2 HCl molecules per A80 molecule? In this case - at IC50 of over 3 uM – the HCl dose is over 6uM. Such a high dose of hydrochloride acid can affect cell growth, survival, stability of certain proteins, etc - and therefore must be controlled by including equimolar levels of HCl to the control arm.

- Likewise, palbociclib is commercially available as palbociclib.HCl or free Palbociclib which is dissolved in HCl/other acid. The authors should provide this information and similarly control for HCl/acidity in the vehicle arm (with A80.2HCl plus Palbociclib having a double dose of HCl)

see PMID: 37230983.

6. In this regard, do the authors see same effects with RNAi to both CDK4 and CDK6 – or with other CDK4/6 inhibitors such as Abemaciclib and Ribociclib.

7. There is no analysis of pRB phosphorylation in response to MYC and KLHL42 over-expression or CDK4/6 inhibition. This information should be provided using phospho-pRB antibodies.

Does KLHL42 induces degradation of hypo- (active; most G1) or hyper- (inactive G2/M) phosphorylated pRB (see issue #3 above)?

8. What is also missing are the reverse experiments. Does over-expression of KLHL42 overcome the effect of MYC plus CDK4/6 inhibitors? Also - are RB-deficient TNBC cells (e.g. BT549 – MDA-MB-436) insensitive to A80.2HCl plus Palbociclib relative to A80.2HCl alone?

Other issues:

- The authors indicate under – “Quantification and statistical analysis. All quantitative data are presented as the means \pm SDs or mean \pm SEM.”

The legends should indicate which method was used in each panel. Alternatively use SD throughout as SEM is smaller/misleading.

- For the A80.2HCl plus Palbociclib combination–the authors should perform a compusyn analysis (<https://www.combosyn.com/>) to determine if the effect is additive or synergistic?

- Model: bottom right – E2F1 should be shown bound to pRb on the chromatin where it acts as a repressor. – The effect may be cytotoxic in some contexts as shown but cytostatic in other contexts.

- Legends are often too succinct / too difficult to follow - and should be carefully expanded /clarified

- Mice were sacrificed when the tumor volume reached 1500 mm³ and were deemed dead.

May change to “Mice reached end-point when....”

Reviewer #3 - CDK4/6i resistance (Remarks to the Author):

CDK4/6 inhibitors define a relatively new class of anti-cancer therapeutics that have been shown to be effective in a subset of breast cancers, with the promise of potential effectivity in other cancer types. The present study rests on the premise that application to other cancer types and understanding mechanisms of intrinsic resistance remain areas of opportunity for discovery. Using a variety of in vitro and in vivo models, the authors identify high MYC expression as a putative driver of resistance, suggest that high MYC reduces the prevalence of the retinoblastoma tumor suppressor protein (RB) through proteasomal degradation, identify KLHL42 as an effector of that degradation who is transcriptional regulation is influenced by MYC, and identify a potential MYC-degrading molecule which could disrupt this process and thereby potentiate the therapeutic advocacy of CDK4/6 inhibitors.

While this is an area worthy of study, and identification of new strategies to improve antitumor response downstream of CDK4/6 inhibitors could be of translational relevance, the study is overall premature, and many of the stated conclusions not sufficiently supported by the data shown. Overall major concerns include the following:

1. A major confounding factor not addressed within the study is the impact of cell cycle position on MYC expression and RB prevalence. Many of the observed changes in protein expression and transcriptional activity can be readily explained by known cell cycle dependent changes in RB and Myc availability and activity.
2. The choice of model systems to assess molecular consequences are not well rationalized, and in many cases do not reflect clinical models in which CDK4/6 would likely be utilized. For example, the prostate cancer model chosen (PC3) is a hormone refractory model reflective of an infrequent variant of prostate cancer which has lost the androgen receptor. In the clinical setting, androgen receptor defective prostate cancers are almost invariably defective already in RB and would be resistant intrinsically to CDK4/6 inhibitors.
3. Assessment of RB presence alone is an insufficient measure of tumor suppressor activity. In order to truly understand the impact on RB function, either in vitro kinase assays can be performed to understand the influence on CDK4/6 activity, or RB phosphorylation status can be assessed alongside total RB. On balance, there is a little confidence that meaningful conclusions can be drawn from the data shown with regard to RB activity.
4. It is unclear why the authors focused on only one member of the RB family, as p107 and 130 may also play a role in cell cycle transitions associated with CDK4/6.
5. The co-immunoprecipitation assays in supplementary figure 3 A cannot be interpreted as described. It is well known that RB activity has already been disrupted in the cell line utilized due to viral oncoproteins.
6. There are no models developed or utilized within the study which can address acquired resistance to CDK4/6 inhibitors. By contrast, multiple studies have been published which developed models of acquired resistance, and concluded highly divergent outcomes from those reported here. Discrepancies should be addressed experimentally.

Authors' Response to Reviewers' Comments on Manuscript NCOMMS-23-28113

We very much thank the Editor and the Reviewers for the positivity and insightful comments for the improvement of our manuscript.

Point-by-point responses to the critiques

Critiques from reviewer #1:

1. In this report, authors first identified MYC as a key to confer cancer cell resistant to CDK4/6 inhibitors. They then found E3 ubiquitin ligase KLHL42 as a transcriptional target of MYC, which induced RB1 degradation. Further, they generated a MYC degrader, A80.2HCl and showed that it worked well to restore RB1 level and reestablished the sensitivity of cancer cells to CDK4/6i. Finally, they demonstrated that combination of CDK4/6i and A80.2HCl could generated better therapeutic effects. The experiments are well planned and performed. The results are clear and supporting the conclusions.

Ans: We thank the reviewer for the positive comments.

2. The identification of A80.2HCl is interesting. The use of this MYC degrader is critical for this paper. However, the information of A80.2HCl is very limited. Is it a chemical or other type of molecule? There is no structure of A80.2HCl provided. Also, it is unclear how it binds to GSPT1 and MYC. Figure 5b is not clear enough and lack of detailed description of the binding information.

Ans: We thank the reviewer for the insightful suggestions. We added the information as reviewer request as following:

- a) A80.2HCl is a chemical molecule screened from imide-based compounds library, we have added the screening information and structural details in the revised **Supplementary Table. 3-5** and **patent application WO 2023/116835 A1**.
- b) We have also provided A80.2HCl compound high-resolution mass spectrometry (**Supplementary Fig. 5**) and NMR data (**Supplementary Fig. 6-8**).
- c) We have also added the detailed binding information between A80.2HCl to MYC and CRBN. In brief, we introduced structural docking studies and demonstrated that A80.2HCl can bind to the dimer interface of MYC and exhibit binding affinity to CRBN (**Fig. 5b** and **Supplementary Fig. 9a**). Specifically, A80.2HCl predominantly binds to a domain consisting of amino acids at positions 324 to 340, a location that aligns with the previously reported binding sites of CRBN for pomalidomide and lenalidomide (*Nature*. 2014 PMID: 25043012) (**Supplementary Fig. 9a**). A80.2HCl can interact with key residues in MYC, including Glu 432, Arg 436 and Arg 439 (**Fig. 5b**).
- d) In addition, we also added the detail of the isothermal titration calorimetry (ITC) assays (**Fig. 5c**) and Homogeneous Time-Resolved Fluorescence (HTRF) analysis (**Fig. 5d**),

which confirmed the direct binding of A80.2HCl to MYC and CRBN in the revised manuscript.

3. The basic information regarding the potential toxicity of A80.2HCl needs to be provided. Did it lead to body weight loss and ALT/AST increase in animal studies?

Ans: We thank the reviewer for the detailed suggestion. We assessed the toxicity of A80.2HCl using BALB/c mice model. In brief, treatment with A80.2HCl did not result in weight loss, liver or kidney injuring by measuring ALT, AST, ALP, and γ -GT levels or histopathological examination (**Supplementary Fig. 11a-c**).

4. In Fig. 5o, A80.2HCl was shown to accumulate within the tumor. Is there any explanation for this tumor-centered accumulation?

Ans: We thank the reviewer for the questions and apologize the misleading. The metabolism and distribution of drugs are closely related to their molecular structure. In the case of A80.2HCl, its chemical structure classifies it as a lipid-soluble drug with a substantial molecular weight. The log P value of A80.2HCl measures at 4.6, indicating its lipophilic nature. These attributes facilitate its enhanced tumor site targeting and prolonged residence within the tumor, as documented in studies on lipid-soluble and molecular size properties of drugs (*Curr Oncol Rep.* 2007 PMID: 17288875; *J Pharm Sci.* 2015 PMID: 25615572). For the explanation of tumor-centered accumulation, it's due to a misunderstanding caused by fluorescent images which the fluorescence intensity threshold was initially set relatively high. To rectify this, we lower the fluorescence intensity and replace the image to present a more accurate representation in the revised **Fig. 5o**.

Reviewer #2:

1. In this manuscript, Ma et al suggest an inverse correlation between MYC and pRB expression (but see below), which confers resistance to the CDK4/6 inhibitor, palbociclib. They subsequently show that MYC induces the Cullin-3-Binding protein KLHL42, which induces ubiquitination and degradation of pRB. Inhibition of MYC reduces KLHL42 expression, increases pRB expression and induces sensitivity to the CDK4/6 inhibitor palbociclib both in vitro and in xenograft assays. The analysis is to the large part well executed and logical, and the results are both interesting and are potentially important therapeutically. However, there are several issues that are of great concern as outlined below each of which can affect the interpretation of results. In addition, the description of the mass spec analysis and the identification of MYC degrader, A80.2HCl, is insufficient - and must be revised.

Ans: We thank the reviewer for the kind comments. As the suggestions, we added the A80.2HCl compound screening information and structural details as shown in the **Supplementary Table. 3-5**. We also provided A80.2HCl compound high-resolution mass

spectrometry (**Supplementary Fig. 5**) and NMR data (**Supplementary Fig. 6-8**).

2. In Supplementary Fig. 1b - the authors show by western blots an inverse correlation between RB and MYC expression. Several lines such as BT549 express no pRb with very high MYC expression - and the assumption/interpretation is that MYC induces degradation of RB via KLHL42. However, at least some if not all of these lines with no discernable pRB expression including BT549 have deletions/truncations in the RB gene – and must not be included in this analysis. (BT549, RB1 c.265_607del1343; MDAMB468, RB1 c.265_2787del12523; DU4475, RB1 c.1_2787del12787; and MDAMB436, RB1 c.607_608ins107; DU145 - RB1 Hom c.2143A>T)

Ans: We agree with the reviewer suggestions and sorry for misleading. We deleted the BT549, DU145, 5637 and 4T1 cells data and repeated this assay with the rest 11 cell lines which have intact RB1 gene. The new result still showed a negative correlation between RB1 and MYC protein expression in the revised **Supplementary Fig. 1a and 1b**.

3. DO the other lines have intact RB1 gene? If MYC or KLHL42 is knocked-down/out in these cells – is pRB expression restored? If not – this figure is misleading.

Ans: We thank the reviewer for the kind comments.

- a) As the reviewer suggestion, we have replaced these data using 11 cell lines with intact RB1 expression (**Supplementary Fig. 1a, b**).
- b) As the request, we also added new data showing that knocking down MYC or KLHL42 restored RB1 protein level in T24, C42 and MDA-MB-231 cell lines (**Supplementary Fig. 3i**). Moreover, knocking down MYC or KLHL42 also restored RB1 protein level in T24 and UMUC14 cell lines (**Fig. 2c, Fig. 3h, Supplementary Fig. 3f and Fig. 4c**).
- c) In addition, while A80.2HCl decreased both MYC and KLHL42 expression, RB1 protein levels were restored in 22RV1, C4-2, MDA-MB-231 and T47D cells (**Supplementary Fig. 9d**).

4. Moreover – this figure shows high expression of MYC plus loss of RB1 (due to deletion/mutations), which contradicts Supplemental Fig. 1a.

Ans: We apologize for misleading the reviewer. In **Supplementary Fig. 1a**, we showed the genetic mutually exclusive between *MYC* amplification and *RB1* deletion, indicating *MYC* amplification maybe involved in RB1 protein regulatory pathway, possibility RB1 protein downregulation in cancer, which inspires us to examine the correlation of protein level between RB1 and MYC. The main concept of our study is that in cancer cells without genetic *RB1* deletion, RB1 loss could still happen in other genetic alterations, such as *MYC* amplification in our finding. We added into the discussion in the revised manuscript (**page**

5, line 102).

5. For Fig 3a – shows FLAG-tagged RB1 immunoprecipitation from 293T cells followed by mass spectrometry analysis to identify RB1-interacting proteins- The authors should provide all the results from this analysis as supplementary excel files as well as Volcano plots with top binders - so one can evaluate what other relevant factors were identified relative to KLHL42.

Ans: We thank the reviewer for suggestions. We have provided those data in the revised manuscript as listed below:

- a) We re-analyzed the mass spectrometry data and uploaded all the analyzed results in **Supplementary. Table 2.**
- b) We also replaced **Fig. 3a and Fig. 3b** with new generated dot plots assay, and data showed the full screening and ubiquitin-associated factors in RB1 immunoprecipitation (**Fig. 3a**).

6. RB protein tends to precipitate and the IP experiments in Fig 3c should carefully control for this with additional antibodies to other E3 ligases or factors that do not bind pRB as negative control.

Ans: We thank the reviewer for the suggestions. We added LRPPRC, RNF40 as the negative control, which is top 3 ubiquitin-associated screening proteins in the revised **Fig. 3a and 3b**.

7. Does KLHL42 degrade other pocket proteins – i.e. p107 or p130?

Ans: Thanks for raising the excellent question. To address the concern, we detected the binding between KLHL42 with RBL1/p107 and RBL2/p130 by immunoprecipitation assays, and our data demonstrated that KLHL42 could not bind with RBL1/p107 or RBL2/p130 (**Fig. 3c**). Knockdown of KLHL42 had little influence on the stability of RBL1/p107 and RBL2/p130. We added those data in the revised **Fig. 3h and Supplementary Fig. 3f**.

8. Does KLHL42 degrade other factors that control the cell cycle that may affect CDK4/6 response?

Ans: We thank the reviewer for raising this great point. To address this issue, we detected the binding between KLHL42 with CDK4 and CDK6 by immunoprecipitation assays. Our data showed that KLHL42 could not bind with CDK4 or CDK6 (**Fig. 3c**). Knocking down KLHL42 also had little influence on the stability of CDK4 or CDK6, we added those data in the revised **Fig. 3c, 3h and Supplementary Fig. 3f**.

9. It was shown that KLHL42 (Cul3/Klhdc5 E3 Ligase) controls microtubule density through its interaction and ubiquitination of p60/katanin (PMID: 19261606). Could

this effect on p60/katanin indirectly affect pRB phosphorylation/degradation? For example, if KLHL42 inhibits cells at G2/M – could this lead to pRB hyperphosphorylation and ubiquitination by other E3 ligases? Can the authors rule out this possibility?

Ans: We thank the reviewer for these helpful suggestions.

- a) To address these concerns, we examined whether p60 influence RB1 status. Our data showed that both the protein and phosphorylation level of RB1 was not significantly changed after knockdown of p60 as shown in **Figure R1** given.
- b) In addition, co-IP assays showed that RB1 had little binding with other E3 ligase such as RNF40 (**Fig. 3b**), and KLHL42 knockdown did not affect the expression of cell cycle-related genes such as CDK4 and CDK6 (**Fig. 3h and Supplementary Fig. 3f**).
- c) Furthermore, we introduced the Glutathione S-transferase (GST) pull-down assay using GST-tagged recombinant RB1 truncations to confirm the direct interaction between RB1 and KLHL42 (**Fig. 3e**).

Based on the above data, we concluded that RB1 is the direct target of KLHL42.

Figure R1. WB analysis from control and p60 knockdown T24 cells.

10. The description of how A80.2HCl was identified is essentially missing in the Methods or text and impossible to understand or follow. The authors should explain exactly how they identified this drug – define GSPT1, CRBN and ITC - and the entire rationale.

Ans: We thank the reviewer for raising these great points. We added the following key information into the revised MS (**page 11, line 273 and line 288; page 12, line 301; page 13, line 338**).

- a) As previously reported, molecules with an imide skeleton, such as pomalidomide and thalidomide, have demonstrated high affinity for E3 ligase cereblon (CRBN) (*Nature*, 2016 PMID: 27338790). We constructed a molecular library of imide-based compounds to screen for drugs able to degrade MYC. This process involved the initial synthesis of 177 distinct imide-based candidates (**Supplementary Table. 3**).
- b) We employed HL60, a MYC drug-sensitive cancer cell line, as the model system. To

this end, we selected the most promising compounds that displayed significant inhibitory effects on cancer cell proliferation for further evaluation of their impact on MYC protein (**Supplementary Table. 4 and 5**). We found that a novel molecule A80.2HCl was capable of completely degrading MYC protein at a concentration of 10nM (**Supplementary Table. 5**).

- c) Using in vitro screening, drug metabolism, pharmacokinetics methods and in vivo validation, we found that A80.2HCl specifically binds to GSPT1 and MYC and subsequently recruits CRBN for degradation (**Fig. 5a and patent application WO 2023/116835 A1**). The further characterizations of A80.2HCl with high-resolution mass spectrometry and NMR analysis are available in the supplementary data (**Supplementary Fig. 5-8**).

11. What other drugs have been identified in this screen? (excel sheet)

Ans: We thank the reviewer for raising this point. We added the 177 compounds screening information in the revised **Supplementary table 3**.

12. What's the structure of A80.2HCl?

Ans: We thank the reviewer for raising this point. We added the A80.2HCl structural details in the revised **Supplementary Table 3 and Supplementary Fig. 5-8**.

13. Is it related to other drugs with similar MYC degradation activity?

e.g. **“A Selective Small-Molecule c-Myc Degradator Potently Regresses Lethal c-Myc Overexpressing Tumors” Adv Sci 2022.**

Ans: We thank the reviewer for the important question. A80.2HCl is a new drug identified by imide-based compounds library, we have provided the screening information (**Supplementary Table. 3-5**) and structural details of A80.2HCl (**Supplementary Fig. 5-8**). We also checked the structure of other reported drugs with MYC degradation activity, our compound is totally different from these drugs.

14. Does A80.2HCl have 2 HCl molecules per A80 molecule? In this case - at IC50 of over 3 uM – the HCl dose is over 6uM. Such a high dose of hydrochloride acid can affect cell growth, survival, stability of certain proteins, etc - and therefore must be controlled by including equimolar levels of HCl to the control arm.

- Likewise, palbociclib is commercially available as palbociclib.HCl or free Palbociclib which is dissolved in HCl/other acid. The authors should provide this information and similarly control for HCl/acidity in the vehicle arm (with A80.2HCl plus Palbociclib having a double dose of HCl) see PMID: 37230983.

Ans: We thank the reviewer for the insightful comments.

- a) Firstly, During the formation of A80.2HCl, hydrochloric acid will dissociate hydrogen ions and chloride ions, and hydrogen ions will be paired to the amino group on the A80 molecule, theoretically, A80.2HCl cannot ionize hydrogen ions to release 2HCl when dissolved.
- b) We detected that the pH values of A80.2HCl and A80.2HCl plus palbociclib solution in DMEM and RPMI-1640 medium that we used. Both the A80.2HCl alone and A80.2HCl plus palbociclib group showed the same >7 pH value instead of <7 PH value, even when A80.2HCl was added up to 10 μ M (**Figure R2** given below).

Figure R2. pH analysis of DMEM and RPMI-1640 medium treated with indicated drugs. data were shown as the mean \pm SD of three independent experiments (n = 3). Two-tailed unpaired Student's t-test. n.s. not significant.

15. In this regard, do the authors see same effects with RNAi to both CDK4 and CDK6 – or with other CDK4/6 inhibitors such as Abemaciclib and Ribociclib.

Ans: We thank the reviewer for these helpful suggestions. As suggested, the efficacy of both abemaciclib and ribociclib in colony formation and cell viability was markedly enhanced as palbociclib did upon addition of A80.2HCl, we added those data in the revised **Supplementary Fig. 13a-c**.

16. There is no analysis of pRB phosphorylation in response to MYC and KLHL42 over-expression or CDK4/6 inhibition. This information should be provided using phospho-pRB antibodies.

Ans: We thank the reviewer for raising this great point. As suggested, our data showed that both phosphorylation RB1 and total RB1 was decreased simultaneously in MYC and KLHL42 overexpressed cells by western blot assay. We provide those data in the revised **Fig. 1f and Supplementary Fig. 12c**.

17. Does KLHL42 induces degradation of hypo- (active; most G1) or hyper- (inactive G2/M) phosphorylated pRB (see issue #3 above)?

Ans: We thank the reviewer for raising these excellent points.

- a) To address this concern, we introduced both hypo-phosphorylated and hyper-phosphorylated RB1 plasmid to examine the KLHL42-induced RB1 degradation. Our data showed that both the hypo-phosphorylated (*Elife*. 2014 PMID: 24876129) (all 15 Ser/Thr Cdk acceptor sites changed to Ala) and hyper-phosphorylated (S245D) RB1 mutants were degraded by KLHL42 expression (**Supplementary Fig. 3d, e**).
- b) In addition, cell cycle analysis revealed that both the RB1 and KLHL42 protein abundance fluctuated during the cell cycle. RB1 protein revealed an inverse correlation with KLHL42 (**Supplementary Fig. 3m, n**). Knockdown of MYC decreased the KLHL42 protein level but increased the both RB1 protein and phosphorylation level in all cell cycle phases (**Supplementary Fig. 3m, n**).

18. What is also missing are the reverse experiments. Does over-expression of KLHL42 overcome the effect of MYC plus CDK4/6 inhibitors? Also - are RB-deficient TNBC cells (e.g. BT549 – MDA-MB-436) insensitive to A80.2HCl plus Palbociclib relative to A80.2HCl alone?

Ans: We thank the reviewer for raising these great points.

- a) As suggested, we examined the effect of MYC plus CDK4/6 inhibitors in KLHL42 overexpressed cells. We demonstrated that KLHL42 overexpression decreased RB1 and dampened the combined effect of MYC plus CDK4/6 inhibitors in both colony formation and cell viability assays (**Supplementary Fig. 12c-g**).
- b) Moreover, the combination treatment of A80.2HCl plus palbociclib did not show significant different from A80.2HCl alone treatment in RB1 deficient BT549 cells (**Supplementary Fig. 12h-j**).

Other issues:

19. The authors indicate under – “Quantification and statistical analysis. All quantitative data are presented as the means \pm SDs or mean \pm SEM.”

The legends should indicate which method was used in each panel. Alternatively use SD throughout as SEM is smaller/misleading.

Ans: We thank the reviewer for mentioning this point. As suggested, the quantitative methods have been indicated in each panel in the revised manuscript.

20. For the A80.2HCl plus Palbociclib combination—the authors should perform a compuSyn analysis (<https://www.combosyn.com/>) to determine if the effect is additive or synergistic?

Ans: We thank the reviewer for raising this great point. As suggested, we confirmed the data that combination of A80.2HCl with palbociclib could synergistically kill T24 and UMUC14 cells by compuSyn analysis mentioned above (*Cancer Res*. 2010 PMID: 20068163), we added those data in the revised **Supplementary Fig. 12a, b**.

21. Model: bottom right – E2F1 should be shown bound to pRb on the chromatin where it acts as a repressor. – The effect may be cytotoxic in some contexts as shown but cytostatic in other contexts.

Ans: We thank the reviewer for mentioning this point. It has now been corrected in the revised manuscript (**Fig. 7**).

22. Legends are often too succinct / too difficult to follow - and should be carefully expanded /clarified

Ans: We thank the reviewer for mentioning this point. It has now been corrected and highlighted in the revised manuscript (**page 30, line 837, 845 and 849; page 31, line 854; page 32, line 864; page 33, line 867, 877 and 879; page 34, line 884 and 888; page 35, line 893 and 903; page 36, line 911 and 915; page 37, line 920 and 926; page 38, line 935; page 39, line 950 and 956; page 40, line 970; page 41, line 975 and 978**).

23. Mice were sacrificed when the tumor volume reached 1500 mm³ and were deemed dead.

May change to “Mice reached end-point when....”).

Ans: Thanks for your suggestions and sorry for misleading. It has been corrected as the reviewer suggested in the revised manuscript (**page 20, line 548page**).

Reviewer #3:

1. CDK4/6 inhibitors define a relatively new class of anti-cancer therapeutics that have been shown to be effective in a subset of breast cancers, with the promise of potential effectivity in other cancer types. The present study rests on the premise that application to other cancer types and understanding mechanisms of intrinsic resistance remain areas of opportunity for discovery. Using a variety of in vitro and in vivo models, the authors identify high MYC expression as a putative driver of resistance, suggest that high MYC reduces the prevalence of the retinoblastoma tumor suppressor protein (RB) through proteasomal degradation, identify KLHL42 as an effector of that degradation who is transcriptional regulation is influenced by MYC, and identify a potential MYC-degrading molecule which could disrupt this process and thereby potentiate the therapeutic advocacy of CDK4/6 inhibitors.

While this is an area worthy of study, and identification of new strategies to improve antitumor response downstream of CDK4/6 inhibitors could be of translational relevance, the study is overall premature, and many of the stated conclusions not sufficiently supported by the data shown.

Ans: We thank the reviewer for the kind comments.

2. A major confounding factor not addressed within the study is the impact of cell cycle position on MYC expression and RB prevalence. Many of the observed changes in protein expression and transcriptional activity can be readily explained by known cell cycle dependent changes in RB and Myc availability and activity.

Ans: We thank the reviewer for raising these excellent points.

- a) To address this concern, we introduced cell cycle analysis of all these proteins. We demonstrated that both the MYC, RB1 and KLHL42 protein abundance fluctuated during the cell cycle. RB1 protein peaked in the G2/M phase but slightly declined in the early G1/S phase as previously reported (*Science*. 2020 PMID: 32703881; *PLoS Biol*. 2017 PMID: 28892491), revealing an inverse correlation with KLHL42 (**Supplementary Fig. 3m, n**).
- b) Moreover, knockdown of MYC decreased the KLHL42 expression but increased both RB1 protein and phosphorylation level in all cell cycle phases (**Supplementary Fig. 3m, n**).

Based on the above, our results showed that although KLHL42 is transcriptional regulated by MYC in cell cycle, KLHL42 induced RB1 degradation is persistent through cell cycle status. We added those data in the revised MS.

3. The choice of model systems to assess molecular consequences are not well rationalized, and in many cases do not reflect clinical models in which CDK4/6 would likely be utilized. For example, the prostate cancer model chosen (PC3) is a hormone refractory model reflective of an infrequent variant of prostate cancer which has lost the androgen receptor. In the clinical setting, androgen receptor defective prostate cancers are almost invariably defective already an RB and would be resistant intrinsically to CDK4/6 inhibitors.

Ans: We thank the reviewer for these helpful suggestions. As suggested by reviewer, we have deleted the AR negative PC-3 cells data and added AR positive C4-2 cells data, and both cell line results showed high MYC could lead to the resistant to CDK4/6 inhibition (**Fig. 1f, g**).

4. Assessment of RB presence alone is an insufficient measure of tumor suppressor activity. In order to truly understand the impact on RB function, either in vitro kinase assays can be performed to understand the influence on CDK4/6 activity, or RB phosphorylation status can be assessed alongside total RB. On balance, there is a little confidence that meaningful conclusions can be drawn from the data shown with regard to RB activity.

Ans: We thank the reviewer for raising these excellent points. To fully address these concerns, we detected the RB phosphorylation status alongside total RB (**Fig. 1f, 3h and**

4c). Indeed, most of the RB phosphorylation status change is consistent with the RB1 protein level change, we added those data into the revised **Fig. 1f, 3h and 4c**.

5. It is unclear why the authors focused on only one member of the RB family, as p107 and 130 may also play a role in cell cycle transitions associated with CDK4/6.

Ans: We thank the reviewer for these helpful suggestions. we detected the binding between KLHL42 with RBL1/p107 and RBL2/p130 by immunoprecipitation assays, and our data demonstrated that KLHL42 could not bind with RBL1/p107 or RBL2/p130 (**Fig. 3c**). Knockdown of KLHL42 had little influence on the stability of RBL1/p107 and RBL2/p130. We added those data in the revised **Fig. 3h and Supplementary Fig. 3f**.

6. The co-immunoprecipitation assays in supplementary figure 3 A cannot be interpreted as described. It is well known that RB activity has already been disrupted in the cell line utilized due to viral oncoproteins.

Ans: We thank the reviewer for mentioning this point. We replaced 293T cell into T24 cells, and the later lack of viral oncoproteins. T24 cell data show RB1 also could bind with KLHL42 (**Supplementary Fig. 3a**). Moreover, we introduced Glutathione S-transferase (GST) pull-down assay using GST-tagged recombinant RB1 truncations and confirmed the direct interaction between RB1 and KLHL42 (**Fig. 3d, e**).

7. There are no models developed or utilized within the study which can address acquired resistance to CDK4/6 inhibitors. By contrast, multiple studies have been published which developed models of acquired resistance, and concluded highly divergent outcomes from those reported here. Discrepancies should be addressed experimentally.

Ans: We thank the reviewer for raising these excellent points.

- a) As the reviewer suggested, we generated the CDK4/6i resistant cell lines, which was obtained by continuously being treated with CDK4/6i (**Supplementary Fig. 14a**). Intriguingly, we found that the RB1 protein levels were dramatically reduced, while MYC and KLHL42 protein abundance was accumulated in resistant daughter cells. Employment of A80.2HCl overcomes the resistance of CDK4/6i caused by MYC and KLHL42 protein accumulation (**Supplementary Fig. 14b**).
- b) Addition of A80.2HCl could sensitize the resistant cancer cells to CDK4/6i treatment implicated by the enhanced cell cycle gene suppression and colony formation inhibition (**Supplementary Fig. 14c-e**).

In conclusion, these above data pointed to MYC could as an attractive target for preventing or overcoming CDK4/6i resistance in cancer cells.

REVIEWERS' COMMENTS

Reviewer #1 (Remarks to the Author)

The authors have nicely addressed the previous review concerns. It is now acceptable for publication in NC.

Reviewer #2 (Remarks to the Author):

The authors addressed my concerns. But a few issues need to be clarified (minor revisions). In additions – the impact of the paper will be improved by professional editing of the manuscript (see below - a few suggestions).

- Regarding point #12

12. What's the structure of A80.2HCl?

Ans: We thank the reviewer for raising this point. We added the A80.2HCl structural details in the revised Supplementary Table 3 and Supplementary Fig. 5-8.

In addition to suppl Table 3, please show the structure of A80.2HCl in Fig. 5 (5b)

Is compound A80 (line 318-321) in Table S3 = A80.2HCl? This may be indicated/highlighted.

Is compound A81 a cis-trans isomer of A80? This and its effect on MYC degradation should be clarified/pointed out in the text. Is A82 also an isomer of of A80?

- Re 17

the hypo-phosphorylated (Elife. 2014 PMID: 24876129) (all 15 Ser/Thr Cdk acceptor sites changed to Ala) RB1 mutants were degraded by KLHL42 expression (Supplementary Fig. 3d, e).

This is a nice and important addition to the paper which should be highlighted/possibly presented in Abstract / primary data -

- The manuscript requires careful editing for English.

For examples:

- Title

RB1 is the gene. The protein is pRB . Change in title (below) and throughout the manuscript.

Induction of KLHL42 by MYC destabilizes pRB and confers resistance to CDK4/6 inhibitor

- ABSTRAT

28 – have shown – show or exhibit

MYC binds to the promoter of the E3 ubiquitin ligase KLHL42 and enhances its transcription, leading to RB1 deficiency by inducing pRB ubiquitination and degradation

We identified a novel compound that degrades MYC, A80.2HCl, which induced MYC degradation when applied at nanomolar concentrations, restored pRB1 protein levels and re-established the sensitivity of MYC high-expressing cancer cells to CDK4/6i.

- Fig. 7 model:

The G1-to-S arrow on the right top is OK. But on the right bottom - to indicate cell cycle progression, the red gradient may be reverse (high left to low on the right).

Line 985: change via to by - responsive to responsible

987: However a novel – change to A novel

989 - sheds light on novel strategies – change to sheds light on potentially novel strategies

Reviewer #3 (Remarks to the Author):

The author has addressed the major concerns of the previous review, and the study has been significantly improved.

Authors' Response to Reviewers' Comments on Manuscript NCOMMS-23-28113A

We very much thank the Editor and the Reviewers for the positivity and insightful comments for the improvement of our manuscript.

Point-by-point responses to the critiques

Reviewer #1 (Remarks to the Author):

The authors have nicely addressed the previous review concerns. It is now acceptable for publication in NC.

Ans: We thank the Reviewer for the positive comments.

Reviewer #2 (Remarks to the Author):

The authors addressed my concerns. But a few issues need to be clarified (minor revisions). In additions – the impact of the paper will be improved by professional editing of the manuscript (see below - a few suggestions).

Ans: We thank the Reviewer for the positivity and insightful comments for the improvement of our manuscript.

1. In addition to suppl Table 3, please show the structure of A80.2HCl in Fig. 5 (5b)

Ans: We thank the Reviewer for the kind comments. The structure of A80.2HCl has now been added in Fig. 5b.

2. Is compound A80 (line 318-321) in Table S3 = A80.2HCl? This may be indicated/highlighted.

Ans: We thank the Reviewer for the helpful suggestions. A80.2HCl represents a specific dosage form designed to enhance solubility based on the A80 compound. This information has now been added and highlighted in the Table S3.

3. Is compound A81 a cis-trans isomer of A80? This and its effect on MYC degradation should be clarified/pointed out in the text. Is A82 also an isomer of A80?

Ans: We thank the Reviewer for the kind comments. A81 is not a cis-trans isomer of A80. The A80, A81, and A82 share an identical molecular formula, with their molecular structures mostly aligned and differed only in the terminal imide ring section, which exists as an enantiomer. As shown in Table S4, both A80, A81, and A82 exhibit notable efficacy against MYC protein. We have now added the information in the Table S3.

4. The hypo-phosphorylated (Elife. 2014 PMID: 24876129) (all 15 Ser/Thr Cdk acceptor sites changed to Ala) RB1 mutants were degraded by KLHL42 expression

(Supplementary Fig. 3d, e). This is a nice and important addition to the paper which should be highlighted/possibly presented in Abstract / primary data

Ans: We thank the Reviewer for the insightful suggestions. We have added the information in the abstract of the revised manuscript **Line 34**.

5. The manuscript requires careful editing for English.

For examples:

- **Title**

RB1 is the gene. The protein is pRB. Change in title (below) and throughout the manuscript.

Ans: We thank the Reviewer for pointing this out. It has been corrected throughout the revised manuscript.

- **ABSTRAT**

28 – have shown – show or exhibit

Ans: We thank the Reviewer for the suggestion. It has been corrected in the revised manuscript **Line 28**.

MYC binds to the promoter of the E3 ubiquitin ligase KLHL42 and enhances its transcription, leading to RB1 deficiency by inducing pRB ubiquitination and degradation

Ans: We thank the Reviewer for pointing this out. It has been corrected in the revised manuscript **Line 32**.

We identified a novel compound that degrades MYC, A80.2HCl, which induced MYC degradation when applied at nanomolar concentrations, restored pRB1 protein levels and re-established the sensitivity of MYC high-expressing cancer cells to CDK4/6i.

Ans: We thank the Reviewer for the kind suggestion. It has been corrected in the revised manuscript **Line 35**.

6. Fig. 7 model:

The G1-to-S arrow on the right top is OK. But on the right bottom - to indicate cell cycle progression, the red gradient may be reverse (high left to low on the right).

Ans: We thank the Reviewer for pointing this out. It has been corrected in the revised manuscript **Fig. 7**.

Line 985: change via to by - responsive to responsible

Ans: We thank the Reviewer for the kind suggestion. It has been corrected in the revised manuscript **Line 1013**.

987: However a novel – change to A novel

Ans: We thank the Reviewer for pointing this out. It has been corrected in the revised manuscript **Line 1014**.

989 - sheds light on novel strategies – change to sheds light on potentially novel strategies

Ans: We thank the Reviewer for the kind suggestion. It has been corrected in the revised manuscript **Line 1017**.

Reviewer #3 (Remarks to the Author):

The author has addressed the major concerns of the previous review, and the study has been significantly improved.

Ans: We thank the Reviewer for the positive comments.